# Seeing Ɔ, remembering C: Illusions in short-term memory

**Marte Otten** [1] *, **Anil K. Seth** [2,3], **Yair Pinto** [1]

**1** Department of Psychology, University of Amsterdam, Amsterdam, The Netherlands, **2** Sussex Centre for Consciousness Science, School of Engineering and Informatics, University of Sussex, Brighton, United Kingdom, **3** Canadian Institute for Advanced Research, Program on Brain, Mind, and Consciousness, Toronto, Ontario, Canada

* M.Otten@uva.nl, marte.otten@gmail.com

**Data Availability Statement:** All relevant data is publicly available from the OSF database (https://osf.io/dfbva/).

**Funding:** AKS is grateful to the European Research Council (ERC-2020-ADG, grant 1010192654) for support.

## Abstract

Perception can be shaped by our expectations, which can lead to perceptual illusions. Similarly, long-term memories can be shaped to fit our expectations, which can generate false memories. However, it is generally assumed that short-term memory for percepts formed just 1 or 2 seconds ago accurately represents the percepts as they were at the time of perception. Here 4 experiments consistently show that within this timeframe, participants go from reliably reporting what *was* there (perceptual inference accurately reflecting the bottom-up input), to erroneously but with high confidence reporting what they *expected* to be there (memory report strongly influenced by top-down expectations). Together, these experiments show that expectations can reshape perceptual representations over short time scales, leading to what we refer to as *short-term memory (STM) illusions*. These illusions appeared when participants saw a memory display which contained real and pseudo-letters (i.e. mirrored letters). Within seconds after the memory display disappeared, high confidence memory errors increased substantially. This increase in errors over time indicates that the high confidence errors do not (purely) result from incorrect perceptual encoding of the memory display. Moreover, high confidence errors occurred mainly for pseudo-to-real letter memories, and much less often for real-to-pseudo-letter memories, indicating that visual similarity is not the primary cause of this memory-bias. Instead 'world knowledge' (e.g., which orientation letters usually have) appear to drive these STM illusions. Our findings support a predictive processing view of the formation and maintenance of memory in which all memory stages, including STM, involve integration of bottom-up memory input with top-down predictions, such that prior expectations can shape memory traces.

## Introduction

Memory is an indispensable part of human cognitive functioning. If we see a new colleague again after a two week vacation, we can usually rely on long term memory (LTM) to put the correct name to the face. If we turn our heads away from a scene, we can rely on our short term memory (STM) to recollect what we just saw. STM is an umbrella term for memory that

**Competing interests:** The authors have declared that no competing interests exist.

stores information for a short period of time (seconds to minutes). Different types of STM have different capacity and time-limits. STM is often hypothesized to consist of short-lived (<500 ms) iconic memory [1] and longer lasting working memory [2]. Sligte and colleagues have additionally suggested the presence of an intermediate form of STM, namely fragile memory [3], which, like WM, lasts at least 4 seconds, but unlike WM has a capacity of 5–15 items. Here, we present 4 studies that explore whether STM in general is susceptible to false memories just like LTM has been shown to be [4, 5]. These experiments specifically focus on a subset of false memories, namely what we call "*memory illusions*": cases were people report with high confidence to have seen something that was not presented to them, i.e., the illusory experience of a reliable and reportable memory. An example of this would be clearly remembering that you put your keys on the dining room table, while in reality you left them on the side table by the front door. Focusing on memory illusions instead of general false memory reports allows a separation of such incorrect but seemingly true memories from memory lapses that are filled in by best guesses (such as when you have no clue where you put your keys, but you check out the dining room table, because you very often drop them there). In such memory lapses, the reliance on pre-existing knowledge is a useful strategy to consciously fill gaps in memory, but it does not reveal whether prior knowledge can actually shape the content of STM. Here, we propose that memories in short term memory are at least partially shaped by world knowledge: prior expectations about the likely events or objects, based on life-long learning and development. In this case, the relevant expectations relate to a highly over-learned category of visual symbols, namely letters of the alphabet.

Recollections of visual stimuli from STM often serve as a proxy for perception [6]. This illustrates the general assumption that STM is a highly reliable storage of recent events: failures to correctly report information that was only recently presented is seldom interpreted as a memory error, but usually seen as a perceptual error. For example, De Gardelle and colleagues [7] showed participants displays which sometimes contained mirrored letters (i.e., a pseudo-letter such as Ɔ). The participants, however, often indicated seeing the real counterpart of the rotated letter (i.e., they report seeing C). De Gardelle and colleagues (2009) [7] conclude that this reflects a perceptual illusion—in other words that their participants make an incorrect per-ceptual inference *at the time of perception*. They did not explore the possibility that their results could be due to memory errors instead of perceptual errors.

However, recall of perceptual information is known to be subject to errors over many time scales. Research in LTM has shown that contextual information that is present both during [8, 9] and after [10, 11] an event can shape the memory for that event, when probed again after a prolonged interval. Arguably, these LTM illusory memories are the result of an adaptive pro-cess, in which memory is an efficient, (re-)constructive system which integrates world knowl-edge within long-term memories to recreate past observations [4, 5]. Indeed, some studies show that world knowledge alone (such as knowledge about social roles) is sufficient to gener-ate memory errors. For example, Kleider and colleagues [12] showed participants faces com-bined with roles (*professor*, *drug dealer*, *artist*). After a 20-minute delay, participants were more likely to incorrectly attribute criminal labels to faces with stereotypically black features. This suggests that an internal bias associating black faces with crime can shape memory. Simi-lar effects were shown for male and female names that were linked to gender-(in)consistent occupations [13]: illusory recollections were more likely to follow gender-based expectations. These results suggest that both external, contextual information and internal expectations about the world can change the content of LTM, resulting in illusory memories.

Here, we asked whether prior knowledge can influence not just LTM, but also STM, result-ing in the generation of illusory short term memories. There is some suggestive evidence that illusory memories may indeed occur in STM: Just 3–4 seconds after reading a list of 3 or 4

interrelated words, participants were more likely to confirm that they saw a semantically related word that was not in the original list, compared to an unrelated new word [14–16]. Interestingly, world knowledge alone may underlie illusory memories in STM. For example, one's cultural experience with music can induce erroneous responses about musical mode and tonality just 1 second after hearing the sequence of tones [17, 18].

Though suggestive, these studies are not yet compelling proof for illusory memories arising in STM, based on knowledge about the world. First, these previous studies did not utilize designs able to distinguish perceptual errors from memory errors. Specifically, if you only test at one single moment after disappearance of the stimuli, then any illusion can either be perceptual (the illusion arose while the participants perceived the stimuli) or memory-based (the illusion arose after perception was finished). One way to resolve this ambiguity is to test participants at (at least) two different moments after stimulus offset. If the frequency of reported illusions increases over time, then at least part of the illusions can be attributed to memory processing rather than perceptual encoding. After all, at both time points initial perceptual processing was finished, so if the illusions arose completely during perception, they ought not to increase over time. In short, characterising the temporal evolution of illusions provides one means of ascertaining whether an illusion is primarily perception-based or memory-based.

Returning to the results of De Gardelle et al. [7], these results show that people report memories of real letters when pseudo-letters were shown. These reports could be the result of illusory memories in which pseudo-letters were progressively replaced with more expected real letters. However, De Gardelle and colleagues interpret their results as perceptual illusions, not memory illusions. Contrarily, other results that are interpreted as STM illusions [7, 14–19] could equally be the result of perceptual illusions that arise when the memory is encoded, after which it is stored and maintained, and subsequently reported, or even just reflect a best guess.

Illusory memories should result in high confidence erroneous responses, where the participants has access to a memory trace (albeit an incorrect memory trace). In the absence of confidence ratings, the erroneous responses could actually consist of low confidence responses or even outright guesses. In these cases, the response is most likely *not* the result of an illusory memory. Instead, it is more likely to result from the absence of a reliable memory, which is compensated by conscious inferences about what is likely to have occurred. Therefore, isolating high confidence responses is necessary to identify those responses that reflect a participant having an actual, albeit false, memory of perceiving something that was not there.

Here we report 4 experiments that test whether STM is susceptible to illusory memories. In all experiments participants see memory displays that contain both real letters, with the standard orientation, and pseudo-letters, which were mirrored. We investigate the occurrence of high-confidence errors (in which participants report seeing a real letter when a pseudo-letter was shown, or vice versa) at different time-points after the display has disappeared. These experiments test whether STM is susceptible to illusory memories, even when the visual information has only just disappeared, and the perceptual memory trace should still be intact [20].

To rule out that these errors reflect a perceptual error that has directly been encoded into STM, we test whether illusory responses increase over time: If the illusions are simply the result of misperceptions, their frequency of occurrence should remain the same over the time-course of STM—since in all cases initial perceptual inference has been completed. Therefore, if the number of reported illusions increases as the retention interval increases, this is best explained by memory-related processes taking place during the retention interval, so during STM storage itself, and not in the perceptual or encoding process. Moreover, we exclude the possibility that the errors are a result of guessing in the absence of reliable memories / reliable perception by

focusing on high confidence errors: errors where the participant explicitly indicates to know the answer, and has not just guessed.

All experiments rely on the extensive experience participants have with the alphabet. Previous experience with letters should have provided participants with abundant evidence that letters typically appear in their standard form. This should provide strong internal expectations about the appearance of the letters of the alphabet. Therefore, occasional instances where the mirrored pseudo-letters are presented might lead observers to incorporate these internal expectations within their memory, and generate illusory memories of true letters, where in fact a pseudo-letter has been shown.

In 4 experiments we thus test 3 hypotheses: 1) Illusions occur in STM: *high confidence* reports of having seen a real letter when a pseudo-letter was shown, or vice versa, occur within 1–2 seconds after a memory display has disappeared. 2) These reports reflect illusory memories, and are not the result of incorrect perceptual encoding: the high confidence illusory reports become more prevalent as memory deteriorates, through the passage of time [21] or through interference [22], and 3) The illusory memories are driven by world knowledge: high confidence illusory reports are more likely to involve pseudo-letter to real-letter errors (as real letters are well-studied and much more prevalent in the real world) than the other way around.

## Experiment 1

### Methods

**Training.**   Prior to the main experiment, participants were trained on a cued change detection task. In this training task, participants were instructed to indicate whether the rectangle at a cued location in a memory display had the same orientation as the rectangle at that same location in a test display. The memory and test displays each consisted of eight white rectangles (1.16˚ × 0.29˚ in size) which had horizontal, vertical or oblique (45˚ or 135˚) orientations. The orientations were pseudo-randomly selected, such that each orientation was equally likely, and that each orientation appeared at least 1, and at most 3 times. The rectangles were presented on a black background, around a red fixation dot (radius: 0.4). Individual rectangles were evenly distributed on an imaginary circle (radius 4.68˚) around fixation. The cue (indicating the target location) consisted of a yellow line that was at one end close (±0.93˚) to fixation and at the other end close to the centre of one rectangle (± 1.6˚). The experiment was programmed in Matlab version 2012b using the Psychophysics Toolbox routines [23]. On each trial the memory display was presented for 0.25 seconds. This was followed by a blank screen of 0.1–1 seconds, a cue for 0.5 seconds and a test display, which remained visible until response. The memory and test display were identical, except that the orientation of the rectangle at the cued location differed 90 degrees between both displays on a randomly selected 50% of the trials. The cue appeared either 100 milliseconds, or 1 second after offset of the memory display. In some trials the cue appeared during a blank interval in between memory display and test display, in other trials the cue appeared together with the test display. In all trials the cue remained visible for 0.5 seconds, and the test display remained visible until response.

Participants could only participate in the actual experiment, if they scored higher than 75% correct (averaged across all trials) on the final 10 blocks of training. We adopted this procedure to ensure that all participants were motivated, and possessed a minimal proficiency at visual memory tasks. See Pinto, Sligte, Shapiro, & Lamme [24] and Sligte, Scholte, & Lamme [3] for a similar training & exclusion procedure. Since only 10% of participants were excluded from

participating in the main experiment based on the training scores, the training procedure was not part of Experiments 2, 3 and 4 in order to not make the overall time on task too long.

**Participants.** 45 participants started training, 5 participants were excluded based on their training results. The remaining 40 participants (28 female; age range 19–30 years, average age 22.2 years) all had normal or corrected-to-normal vision. All participants were naïve to the purpose of this experiment. All participants gave their written informed consent to participate in the study, which was approved by the local ethics review board of the University of Amsterdam. The participants participated for monetary compensation (10 euro per hour) or student credits.

**Stimuli.** The memory and cue displays were set up in the same way as the displays in the training session, with 6 or 8 letters replacing the tilted bars. Pseudo-letter were mirrored (i.e. flipped from left to right). Therefore only letters that changed when mirrored were included in the experiment (i.e. the symmetrical letters A,H,I,M,O,T,U,V,W,X,Y were excluded from the memory and response displays). When the memory display consisted of 8 letters, it contained 2 pseudo-letters. When it contained 6 letters, it had a 50% chance of containing 1 pseudo-letter and a 50% chance of containing 2 pseudo-letters.

To increase memory decay through task-irrelevant interference, a second display was included in each trial. This display consisted of randomly selected letters, unrelated to the memory display. Furthermore, it contained the same proportion of real and pseudo-letters as the memory display. Participants were instructed to ignore this second display.

The response display consisted of 6 letters: the target letter (with the same orientation as in the memory display, or mirror reversed), two other randomly selected letters from the memory display, and 3 letters that did not appear in the memory display.

The target letter was a real letter in 75% of the trials and a pseudo-letter in the remaining 25%. If the target was a pseudo-letter, then it appeared as a response option randomly on half of those trials while on the other half of pseudo-letter trials only the real counterpart was one of the response options. If the target was a real letter, then it appeared as a response option on a randomly selected 83% of the trials, while its mirror reversed (pseudo-letter) version appeared as a response option on the other 16% of the trials. This ensured that of the critical trials (trials where participants could exhibit memory illusion: trials in which a real- or pseudo-letter target was combined with its mirrored counterpart in the response screen) about half were real-to-pseudo trials (approximately 13% of all trials), while the other half were pseudo-to-real trials (again, approximately 13% of all trials). Together, the critical trials made up ~220 trials, 26% of all trials in the experiment. It is important to note, though, that throughout the experiment participants were more likely to encounter real letters as the target, and as the correct response option, than pseudo-letters, in line with the overall expectation that real letters are the standard letter-form.

**Procedure.** The experiment consisted of 2 sessions of 18 blocks of 24 trials, see Fig 1. The first session started with a practice block of 12 trials (this block was not included in the analysis). The experiment had three memory conditions. In two conditions the memory probe appeared 0.75 seconds after offset of the memory display, in the third memory condition, the probe appeared 3 seconds after offset of the memory display. If the probe appeared 0.75 seconds after memory display offset, it either appeared together with the irrelevant second display (the Short-with-interference condition), or it appeared in isolation (the Short condition). If the probe was presented 3 seconds after offset of the memory display (the Long condition), it appeared in isolation. If the memory probe appeared in isolation (the Short and Long conditions), the irrelevant second display appeared 5 seconds after memory display offset, which meant that the interval between the probe and the irrelevant second display was either presented for 1.5s (Long delay) or 3.75s (short delay). In all cases, the memory probe was shown

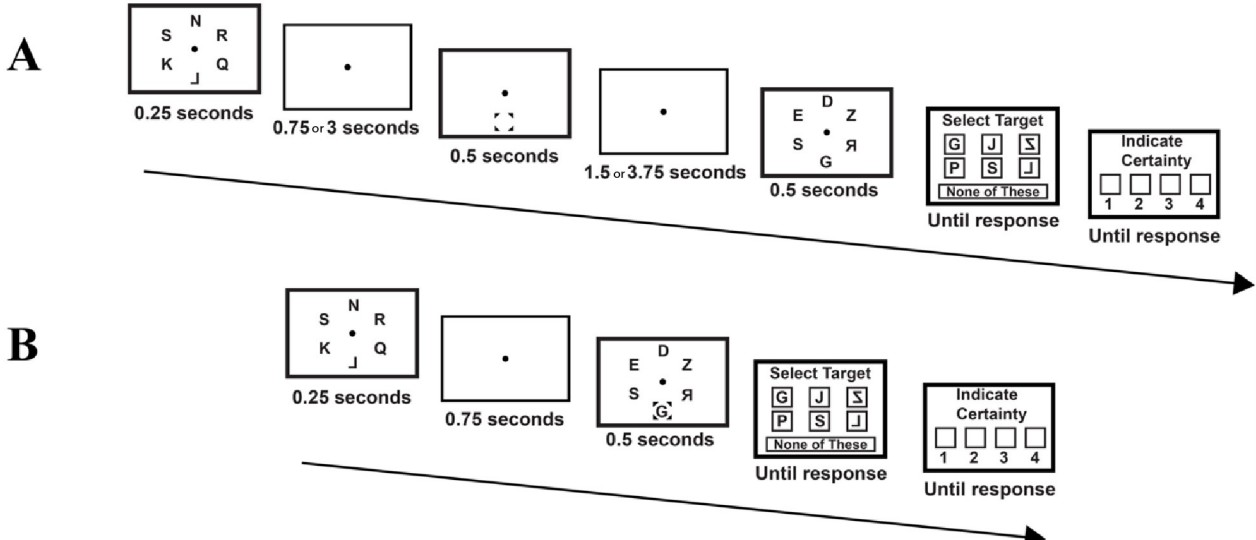

**Fig 1. Experiment 1—Procedure.** The memory display consisted of 6 or 8 letters. It contained 1 to 2 pseudo letters (set size 6) or 2 pseudo letters (set size 8). In all 3 memory delay conditions the trial started with the memory display appearing for 0.25 seconds. Subsequently in the Short and Long conditions (panel A) the memory cue appeared after a blank of 0.75 seconds (Short) or 3 seconds (Long), for 0.5 seconds, on a blank screen. Subsequently, 5 seconds after offset of the memory display, an irrelevant second letter display, containing the same amount of (randomly selected) real and pseudo-letters as the memory display, appeared for 0.5 seconds. The Short with Interference condition (panel B) was the same as the Short condition, except that now the second letter display appeared together with the cue. In all conditions, after offset of the second letter display, the participant first indicated the identity of the letter at the cued location in the memory display. Subsequently she indicated confidence in her judgment (1 = guess, 4 = certainty).

for 0.5 seconds and the irrelevant second display was presented for 0.5 seconds. The three trial-types were randomly intermixed.

Participants were instructed to report the letter that appeared at the cued location. They were explicitly instructed not to confuse real and pseudo-letters and received feedback on their performance after each trial. They were instructed to select "none of these" if the target letter was not a response option. After selecting the target letter, participants indicated their confidence by clicking on one of four boxes (from 1–4, 1 = guess, 4 = certain).

**Analysis.** For each memory delay (short, short with interference and long) and each target type (pseudo- and real letter targets) the occurrence of illusory errors (real-to-pseudo or pseudo-to-real errors) and other errors (selection of one of the unrelated distractor items) was calculated.

Exclusion of participants with too few high confidence critical trials in one or more cells of the design (10 trials or less) were excluded from the analysis resulted in an inclusion of 23 participants (17 participants were excluded) when the results were averaged across the set size conditions. Included participants had on average 53.8% high confidence trials over all critical conditions (range: 36.3–93.2, sd: 13.9). Excluded participants had on average 26.2% high confidence trials over all conditions (range 3.6–47.7%, sd: 13.0).

The resulting data were analysed in a Bayesian repeated measures ANOVA in JASP 0.16.4 [25], with the factors Target Type (real or pseudo-letter), Memory Delay (short, short with interference and long) and Error Type. Within this analysis, the posterior odds of all different combinations the three factors and their interactions are compared to the prior explanatory value of each model. Based on this analysis, we selected the model with the best fit, reporting the BFm. Individual model components (main effects and interactions) were additionally evaluated based on their BFincl across all models. Bayes Factors were interpreted as follows:

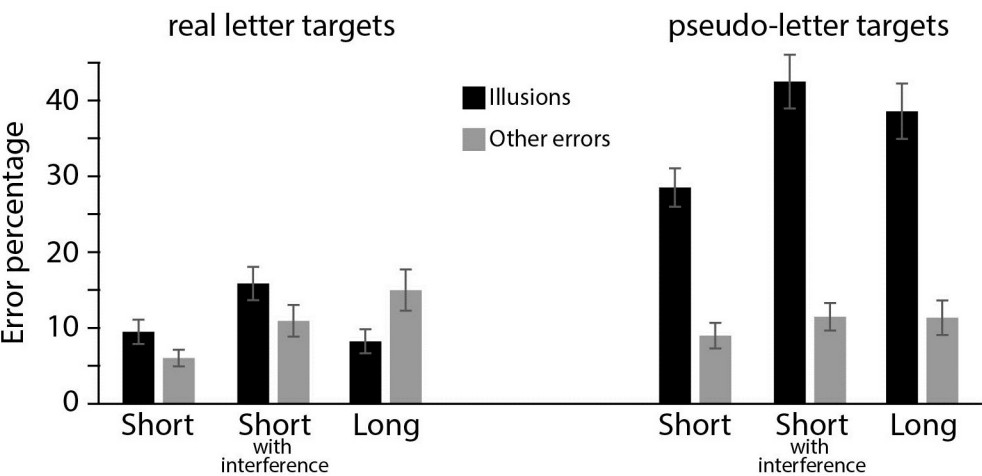

**Fig 2. Experiment 1.** Error rates in trials where participants indicated high confidence in their response. Percentages of expectation-induced errors and other errors are depicted for pseudo-letter targets (right display) or a real letter targets (left display). All error rates are reported for each of the three Memory Delay conditions.

BF < 0.1: strong evidence for the null hypothesis. 0.1–0.33: substantial evidence for the null hypothesis. 0.33–1: anecdotal evidence for the null hypothesis. 1–3: anecdotal evidence for the alternative hypothesis, 3–10: substantial evidence for the alternative hypothesis, >10: strong evidence for the alternative hypothesis [26].

We investigated our three main hypotheses: hypothesis 1–do illusions occur in STM?—by determining if participants confused pseudo-letter targets with their real counterpart with high confidence, or the other way around, more often than other high confidence errors (main effect of Error Type). We tested our second hypothesis—do illusions increase as memory deteriorates?—by measuring if illusions increased from the Short to the other memory conditions (interaction between Memory Delay and Error Type). Finally, we tested our third hypothesis—does world knowledge drive STM illusions?—by investigating if high confidence errors confusing pseudo-letter targets for their real counterparts occurred more often than the other way around (Target Type interacts with Error Type).

## Results

Fig 2 shows the illusory and other error rates in high confidence trials. The Bayesian repeated measures ANOVA showed that a model with main effects for Error Type, Target Type, and Memory Delay, and a Target Type * Error Type and Memory Delay * Error Type interaction had the best fit (BFm = 18, see S1 Appendix for full model comparison and Table 1 for an overview of BFincl for each model component).

**Table 1. Error rates ANOVA.**

| Effects | P(incl) | P(excl) | P(incl\|data) | P(excl\|data) | $BF_{incl}$ |
|---|---|---|---|---|---|
| Target Type | 0.737 | 0.263 | 1.000 | $2.442^{e-15}$ | $1.462^{e+14}$ |
| Memory Delay | 0.737 | 0.263 | 0.999 | $8.189^{e-4}$ | 435.745 |
| Target Type*Memory Delay | 0.316 | 0.684 | 0.189 | 0.811 | 0.504 |
| Error Type | 0.737 | 0.263 | 1.000 | $2.442^{e-15}$ | $1.462^{e+14}$ |
| Target Type*Error Type | 0.316 | 0.684 | 1.000 | $2.442^{e-15}$ | $8.871^{e+14}$ |
| Memory Delay*Error Type | 0.316 | 0.684 | 0.660 | 0.340 | 4.205 |
| Target Type*Memory*Error Type | 0.053 | 0.947 | 0.120 | 0.880 | 2.445 |

Hypothesis 1 was supported: Illusions (pseudo-to-real or real-to-pseudo) were much more prevalent than other errors (a main effect of Error Type, BFincl>1000). The interaction between Error Type and Memory Delay, (BFincl = 4.21) shows support for hypothesis 2: as memory deteriorates with a larger memory delay or more visual interference, the increase in illusory errors (7% increase from 19% in the short delay to 26% averaged over the short with interference and long delay) is larger than other errors (4% increase, from 8 to 12%). The interaction between Target Type and Error Type (BFincl >1000) provides evidence for Hypothesis 3: when the target is a pseudo-letter, illusions are very common (37%) but other errors occur much less often (11%). For real letter targets, the illusory error rate is the same as the rate of erroneously selecting any other item from the response display (11% in both cases). Finally, there is only anecdotal evidence for a three-way interaction between Target Type, Error Type and Memory Delay (BFincl = 2.45): For pseudo-letter targets, illusory errors increase 12% as memory deteriorates through interference of time or visual information (29% illusions to 41% illusions). None of the other conditions are as sensitive for memory decay, with other errors for pseudo-letter targets increasing 3%, and illusory errors and other errors for real letter targets increasing 2% and 7%, respectively.

## Discussion Experiment 1

In Experiment 1 we observed an abundance of pseudo-to-real letter memory illusions. In fact, when the probe was after an interfering visual display, high confidence pseudo-to-real illusions (43%) were almost as prevalent as high confidence selections of the actual pseudo-target (50%). That is, when participants were sure of their memory, the recollection was almost as likely to be illusory as it was to be real.

What causes these illusory reports? One option is incorrect perception or encoding when the memory is formed. However, Experiment 1 shows that expectation-induced memory illusions increase as memory declines. From the Short (memory probe 0.75 seconds after offset of the memory display with no visual interference) to the Long condition (memory probe after 3 seconds) or the Short with interference (memory probe after .75 seconds following an interfering display) the prevalence of high confidence expectation induced illusions increased from 29% to 39% and 43%, respectively. This indicates that high confidence errors are not purely the result of errors in perception or encoding: when probed 0.75 seconds after the memory display has disappeared, people correctly report what they saw in 70% of high confidence trials, thus showing that initial perceptual encoding was relatively intact. When the memory trace deteriorates, high confidence reports are moving towards an equal division between accurate and illusory memories. This effect must be due to processes ongoing in STM: Had these illusory memories been the result of changes in perception, they should have been present at the earlier probe as well.

Another cause of illusory reports could be confusion between the to-be-remembered target and the mirrored counterpart that is presented in the response screen. However, Experiment 1 shows that pseudo-to-real illusions were much more prevalent than real-to-pseudo illusions. This suggests that world knowledge is a more important driver of illusory memories than visual similarity, as visual similarity between to-be-remembered target and the mirrored counterpart that is presented in the response screen is clearly identical in both cases.

Yet another reason why illusory reports arise could be a response bias: Participant might just be more inclined to select real letters instead of pseudo-letters, since real letters are a well-known category. If the illusory reports are actually not illusions, but due to response bias, then the pattern of responses that can be attributed to a response bias (more expectation-induced illusions [Ɔ to C] than similarity-induced illusions [C to Ɔ]) should be present not only for the

high-confidence responses; indeed, it should be even stronger for the low-confidence responses. This is because a response bias is present whether the participant thinks they remembered correctly or not, and a participant ought to rely more heavily on a response bias when they have only a weak memory of what they saw. However, in Experiment 1 the tendency to report more expectation-induced illusions than similarity induced illusions is clearly present in high confidence trials, but is much less pronounced in low confidence trials—an opposite finding to what one would expect if these effects were the result of a response bias: On high confidence trials pseudo-letter targets were more often confused for their real counterparts (36%) than on low confidence trials (22%). However, the reverse was observed when real letter targets were confused for their pseudo counterpart (high confidence: 11%, low confidence: 17%).

This also shows that the current illusory memories are quite different from earlier reports of swap errors in STM [27–30], where participants incorrectly report having seen features of a distractor, instead of the target feature. While the currently observed memory illusions are more likely to occur in high confidence trials than on low confidence trials, swap errors generally occur on low confidence trials instead of high confidence trials [28].

## Experiment 2

The illusory memories observed in Experiment 1 seem to be driven by world knowledge, i.e. long-standing knowledge about the usual orientation of letters. In Experiment 2, we tested whether transiently induced expectations can also cause illusory memories in STM. To this end, we associated the color of the fixation cross with either a high or a low probability of a pseudo-letter target. If transiently induced expectations can induce illusory memories, then pseudo-to-real illusions should be more prevalent when the fixation color induced the expectation of a real letter target, than when it induces the expectation of a pseudo-letter target.

Experiment 1 also had different proportions of trials where the real letter counterpart of the target was presented in the probe display, compared to the pseudo-letter counterpart. Although the results of Experiment 1 (with its marked increase in illusory memories when memory decays) suggest that this difference in occurrence of the mirrored (pseudo) counterpart in the response screen over the entire experiment is not the cause of the memory illusions we observe, it is important to completely rule this out. Therefore, the response screen in Experiment 2 always included both the target and its mirrored counterpart. This equates the presence of real letter and pseudo-letter probes in the response display over the entire experiment, and makes sure that the participant cannot become biased to select one of the two options simply because it is generally more prevalent in the response screen.

### Methods

**Participants.**   In this experiment, 290 participants took part (164 female; age range 18–56 years, average age 20.3 years), all having normal or corrected-to-normal vision. All participants were naïve to the purpose of this experiment and did not take part in Experiment 1. All participants gave their written informed consent to participate in the study, which was approved by the local ethics review board of the University of Amsterdam. The participants participated for student credits as part of a course within the first-year Psychology curriculum. The experiment was part of a 2-hour session which contained this experiment, and several other unrelated experiments. The number of participants was determined not by the experimenter but by student enrollment in the course, which explains the large increase in participants compared to Experiment 1.

**Stimuli.** The memory display consisted of 6 letters, all with different identities, and a fixation cross (two filled intersecting rectangles, with a size of 0.5˚ x 0.1˚). 2 of the 6 letters in the memory display were pseudo-letters. The second, irrelevant, display also consisted of 4 real and 2 pseudo-letters. The letters in the second display were randomly selected and unrelated to the letters in the memory display. A cue highlighted one of the locations previously occupied by one of the presented letters. This cue consisted of four filled triangles (size of each triangle: width: 0.3˚, height: 0.3˚, shape: two orthogonal lines connected by a 45˚ line), arranged to outline a square (of 1.4˚ x 1.4˚). The response screen consisted of 6 letters, 3 real letters and their pseudo counterparts. 2 pairs of letter/pseudo-letters had been part of the memory display (either as real or as pseudo-letter) and 1 pair had not been part of the memory display. The letter at the cued location was always presented in the response display, additionally one of the other letters presented in the memory display was also one of the response options. Since the target letter (and its counterpart) were always among the response options, the response option "none of these" was eliminated. Furthermore, both the central fixation cross and the cue were either blue (CIE: 0.148, 0.078, luminance: 6.47 cd/m$^2$) or red (CIE: 0.641, 0.341; luminance: 11.9 cd/m$^2$).

**Procedure.** Before the actual experiment each participant received instructions about the goal of the task, namely to correctly report the cued letter from the first display. The participant was extensively instructed to not confuse real and pseudo-letters. Furthermore, participants performed 12 practice trials. The practice trials were identical to the experimental trials, except that the memory display was shown for 2.5 seconds instead of 250 ms. After each trial the participant received feedback (correct/ incorrect response) on her performance. The main experiment consisted of 12 blocks of 20 trials. After each block the participant received feedback on her performance which included information about her average accuracy. This extensive feedback was provided to keep the participants motivated. See Fig 3 for an overview of the procedure.

Each trial started with the presentation of a fixation cross on an otherwise blank screen for 0.5 seconds. After this, the memory display was presented for 0.25 seconds, a blank screen, only containing the fixation cross was presented for 0.5 (the Short condition) or 3 seconds (the Long condition). Then the cue was presented for 0.25 seconds, again followed by a fixation cross on a blank background. 3.75 seconds after the offset of the memory display, the second display was presented for 0.25 seconds (participants were instructed to ignore this second display). After this the participant first chose the target letter from 6 different response options, and then indicated her confidence (on a scale from 1–4). The color of the fixation cross was the same throughout the entire trial, but varied across trials. The color of the fixation cross and the memory cue were identical. If the fixation cross and the cue were red, a pseudo-letter was the target on 25% of the trials. The target was a pseudo-letter on 60% of the trials when the

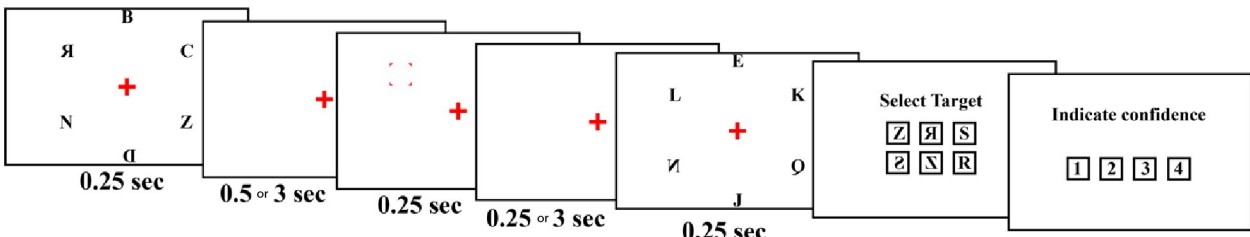

**Fig 3. An overview of the procedure of Experiment 2.** Each trial started with a fixation cross for 0.5 seconds (not depicted), followed by the depicted sequence of events. The final two response displays remained visible until the participant had indicated her choice.

fixation cross and the cue were blue. Participants filled out a brief questionnaire in which they were invited to indicate what the experiment was about after the experiment. The relationship between the colour of the cue and the nature of the target was not mentioned in any of the responses. This experiment contained two memory delay conditions (short and long), two target types (real and pseudo) and two fixation color conditions (red and blue). All conditions were randomly intermixed throughout the experiment.

**Analysis.** First, we tested whether the cue colour influenced the occurrence of illusory errors, analysing the illusory error rates (real-to-pseudo or pseudo-to-real errors) in an ANOVA with target type (real or pseudo letter) and cue colour (red or blue, indicating low and high probability of a pseudoletter target) as factors. Since this analysis showed no effects of cue colour (see Results), the rest of the analysis was conducted while collapsing over the two cue colour conditions. As in Experiment 1, error rates on high confidence trials were analyzed in an ANOVA with target type (real or pseudo letter), memory delay (long or short) and error type (illusions -from pseudo-to-real or the other way around- vs other errors) as independent variables and relative occurrence of high confidence selection as the dependent variable. Exclusion of participants with too few high confidence trials in one or more cells of the design (10 trials or less) resulted in an inclusion of 198 participants (92 participants were excluded) when the results were averaged across the two color conditions. Included participants had on average 49.2% high confidence trial over all conditions (range: 24.6–95.8, sd: 13.3). Excluded participants had on average 15.7% high confidence trials over all conditions (range 0.4–38.8%, sd: 9.5).

The repeated measures ANOVA allowed a direct test whether the results of Experiment 1 were still observed when the probe display contained both the target and its mirrored (pseudo) counterpart. Hypothesis 1 (Illusory memories occur in STM) would be supported if expectation-induced errors are more prevalent than other errors, resulting in a main effect of Error Type. Hypothesis 2 (These illusions are memory-driven) would be supported if high confidence illusory reports become more prevalent with longer memory delays, resulting in an interaction between Error Type and Memory Delay. Hypothesis 3 (The illusory memories are driven by expectations about the world) would be supported if the high confidence illusory reports are more likely to occur when the target in the memory display is a pseudo-letter compared to a real letter, resulting in an interaction between Target Type and Error Type.

The additional hypothesis, for this Experiment, is that locally induced expectations can also lead to expectation-induced memory illusions. This hypothesis would be supported if cue colour influences the occurrence of memory illusions: Specifically, more pseudo-to-real memory illusions should occur after a red fixation cross (indicating a higher probability of a real letter target) than after a blue fixation cross.

## Results

First, we tested whether cue colour, and thus implicit expectations about the likelihood of real versus pseudo-letter targets, changed the number of illusory errors the participants made for real and pseudo-letter targets. The Bayesian repeated measures ANOVA showed that the best model only contained a main effect of target type (BFm = 38.8, BFincl > 1000). There was strong evidence to not include a main effect of cue colour (BFincl = 0.07) nor the interaction between cue colour and target type (BFincl = 0.06). Because of this, in subsequent analyses all data are collapsed over the two cue types (red and blue).

Fig 4 shows the memory illusions and other error rates for high confidence trials, depicted per target type and memory delay. We tested whether the results of Experiment 1 were replicated when the probe display contained both the target that was presented in the memory

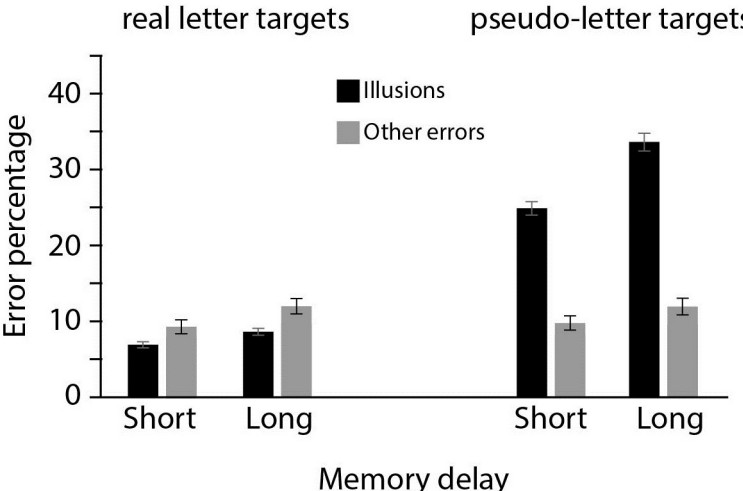

**Fig 4. Experiment 2.** Error rates in trials where participants indicated high confidence in their response. Percentages of expectation-induced errors and other errors are depicted for pseudo-letter targets (right display) or real letter targets (left display). All error rates are reported for the short and long Memory Delay conditions.

display, and the mirrored (pseudo) counterpart. The Bayesian repeated measures ANOVA showed that the full model, with main effects for error type, target type, and memory delay, all two-way interactions and the three-way interaction had the best fit (BFm > 1000, see S1 Appendix for full model comparison and Table 2 for an overview of BFincl for each model component). Participants were more likely to select the mirrored (pseudo) counterpart of a target than to commit other types of errors (18% vs 11%, main effect of error type, BFinlc>1000) and this effect of error-type interacted with the type of target (BFincl>1000): pseudo-letter targets were more likely than real-letter targets to evoke an expectation-induced illusory response (going from 8% illusions for real-letter targets to 29% illusions for pseudo-letter targets), while the other (non-illusory) error rate was not influenced by target-type (11% for real-letter and pseudo-letter targets).

Memory delay had a clear effect on error rates, which depended on both error type and target type, resulting in a three-way interaction between Memory Delay, Error Type and Target Type (BFincl = 206). Follow-up ANOVAs conducted separately for each error type showed that for illusory errors, the optimal model consisted of the factors Memory delay, Target type and the interaction between the two (BFm>1000, see S1 Appendix for full model comparison and Table 3 for an overview of BFincl for each model component). Pseudo-letter targets were more likely to evoke memory illusions than real-letter targets (BFincl >1000 for Target type),

**Table 2. Error rates ANOVAs.**

| Effects | P(incl) | P(excl) | P(incl\|data) | P(excl\|data) | BF$_{incl}$ |
|---|---|---|---|---|---|
| Target Type | 0.737 | 0.263 | 1.000 | 0.000 | ∞ |
| Memory Delay | 0.737 | 0.263 | 1.000 | $6.62^{e-12}$ | $5.38^{e+10}$ |
| Target Type*Memory Delay | 0.316 | 0.684 | 0.992 | 0.008 | 259.97 |
| Error type | 0.737 | 0.263 | 1.000 | 0.000 | ∞ |
| Target Type*Error type | 0.316 | 0.684 | 1.000 | 0.000 | ∞ |
| Memory Delay*Error type | 0.316 | 0.684 | 0.972 | 0.028 | 73.87 |
| Target Type*Memory Delay*Error type | 0.053 | 0.947 | 0.920 | 0.080 | 206.37 |

**Table 3. Follow-up ANOVA for illusory error-rates.**

| Effects | P(incl) | P(excl) | P(incl\|data) | P(excl\|data) | BF$_{incl}$ |
|---|---|---|---|---|---|
| Memory Delay | 0.600 | 0.400 | 1.000 | 0.000 | $\infty$ |
| Target Type | 0.600 | 0.400 | 1.000 | 1.699$^{e-13}$ | 3.925$^{e+12}$ |
| Memory Delay ✱ Target Type | 0.200 | 0.800 | 1.000 | 1.521$^{e-4}$ | 26289 |

longer delays resulted in more illusory errors (BFincl >1000 for Memory Delay). The interaction between Memory delay and Target type (BFincl > 1000) shows that increasing the memory delay had different effects on the frequency of memory illusions depending on the type of target letter. With pseudo-letter targets, a longer memory delay (from Short to Long) generated an 8% increase of expectation-induced memory illusions (from 25% to 33%). With real letter targets expectation-induced illusions barely increased from short to long delays (2% increase, from 7% to 9%).

For other, non-illusory errors only Memory Delay was a relevant factor (BFm = 42, see S1 Appendix for a full model comparison and Table 4 for an overview of BFincl for each model component). Non-illusory error rates increased from the short to the long delay (BFincl = 190 for Memory Delay) but this increase was the same for pseudo- and real letter targets (2% increase, from 10 to 12%, in both real and pseudo-letter targets).

## Discussion

Experiment 2 confirms the findings of Experiment 1 that expectation-based illusory memories in STM exist, and occur even when the memory display has only just disappeared from view. As in Experiment 1, Experiment 2 clearly shows that, as memory declines, memory illusions become more prevalent, suggesting that the role of internal expectations in shaping memory reports increases. The results of Experiment 2 moreover clearly indicate that the observed memory illusions are based on world knowledge, and not just on visual similarities between the target and the chosen mirrored counterpart. As in Experiment 1, illusions evoked by a pseudo-letter are more prevalent than illusions induced by a real letter, even when both the pseudo-letter and the real letter are simultaneously present in the response display. Together, these findings suggest that memory of very recent events is promptly shaped by world knowledge, and that the effect of world knowledge increases as time passes and the memory trace itself decays.

However, transiently generated expectations, in this case provided by a cue about the prevalence of real versus pseudo-letters targets, had no effect on the prevalence of illusory memories. This is consistent with the notion that expectations need substantial time to develop before they can affect explicit, high confidence memories.

## Experiment 3

In Experiments 1 and 2, real letters occurred more often in the memory and probe displays than pseudo-letters, and thus real letters were more likely to be targets than pseudo-letters,

**Table 4. Follow-up ANOVA for other, non-illusory error-rates.**

| Effects | P(incl) | P(excl) | P(incl\|data) | P(excl\|data) | BF$_{incl}$ |
|---|---|---|---|---|---|
| Memory Delay | 0.600 | 0.400 | 0.087 | 0.913 | 0.064 |
| Target Type | 0.600 | 0.400 | 1.000 | 1.705e-5 | 39110.895 |
| Memory Delay ✱ Target Type | 0.200 | 0.800 | 0.009 | 0.991 | 0.035 |

with pseudo-letters making up 25% (Experiment 1) and 33% (Experiment 2) of the targets, while real letters were the target in the remainder of trials. One might worry that a larger proportion of targets compared to non-targets within the total stimulus-set could bias participants to indicate that they had seen the target (Criss, 2010) [46]. It is possible that the same mechanism applies when a specific target (in this case real letters) is more frequent within the stimulus set than another target type (in this case pseudo-letters). This, too, could bias participants to respond that they have seen a real letter when they have in fact seen a pseudo-letter. In this case, the larger proportion of pseudo-to-real responses relative to real-to-pseudo responses would not be the result of a lifelong exposure to the alphabet, but instead an experiment-long exposure to the stimulus sets used in Experiments 1 and 2. To test this, Experiment 3 uses a completely balanced stimulus set: pseudo-letters make up half of each memory display, and are cued as targets in half of the trials.

Experiment 1 and 2 consistently show that participants are more inclined to mistakenly report having seen the real counterpart of a pseudo-letter target, and that these pseudo-to-real illusions are more likely than reporting real-to-pseudo illusions. If this is caused by the unbalanced stimulus sets, then Experiment 3 should show that participants exposed to a balanced stimulus set are equally likely to confuse a pseudo-letter with its real counterpart as the other way around. If, however, the higher incidence of pseudo-to-real letter illusions results from world knowledge that real letters are more common than pseudo-letters, then this difference should be present also when the stimulus set itself is completely balanced with regards to the presence of pseudo- and real letter(-target)s.

## Methods

**Participants.** In this experiment, 64 participants participated (51 women; age range 17–46 years, average age 20.5 years), all having normal or corrected-to-normal vision. All participants were naïve to the purpose of this experiment. All participants gave their written informed consent to participate in the study, which was approved by the local ethics review board of the University of Amsterdam. The participants participated for student credits within the first-year Psychology curriculum. Note that in this experiment the number of participants was determined based on a stopping rule implemented in an ongoing sequential analysis of the Bayes Factor for the critical t-test, namely the comparison between the effect of Error-Type for Real and Pseudo-letter targets. Every other day of testing, the sequential analysis was run, and once the outcome crossed a BF of 10, testing would stop.

**Stimuli.** The memory displays and irrelevant intervening screens were the same as in Experiment 2 except that the memory displays always consisted of 3 real and 3 pseudo-letters.

**Procedure.** The experiment consisted of 18 blocks of 25 trials. The procedure was identical to Experiment 2, with two exceptions. First, the time between memory display and memory probe was always 3 seconds (i.e., the Short condition was eliminated). Secondly, the display time for the memory display in practice trials was variable (where it was 2500 ms in for all practice trials in Experiment 2: In the first practice trials the memory display was presented for 1350 ms. In each subsequent practice trial, the presentation of the memory display was shortened by 100 ms, so that the final practice display had the standard presentation time of 250 ms.

**Analysis.** Exclusion of participants with too few high confidence trials in one or more cells of the design (10 trials or less) resulted in an inclusion of 59 participants (5 participants were excluded). Included participants had on average 38.9% high confidence trial over all conditions (range: 8.4–64.4, sd: 14.6). Excluded participants had on average 2.8% high confidence trials over all conditions (range 0.9–4.4%, sd: 1.7).

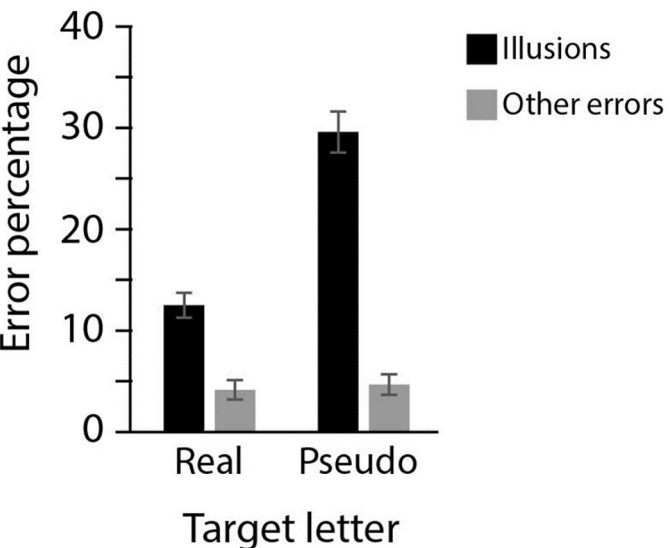

**Fig 5. Experiment 3.** Error rates in trials where participants indicated high confidence in their response. Percentages of expectation-induced errors and other errors are depicted for pseudo-letter and real letter targets.

The relative occurrence of illusory memories (on high confidence trials) was analyzed in an ANOVA with target type (real or pseudo letter), and type of error (expectation induced illusory error—from pseudo-to-real or vice versa—or other error) as independent variables. Because of listwise deletion of missing values (resulting from 0 high confidence responses in one or more of the cells of the design), one participant was excluded from the main ANOVA.

The aim of this experiment was to test whether illusory memories based on expectations about the world still arise when the stimulus set contains the same proportion of familiar stimuli (real letters) as novel stimuli (pseudo-letters). Therefore, only Hypothesis 1 (Illusory memories occur in STM) and Hypothesis 3 (The illusory memories are driven by expectations about the world) are tested. These hypotheses would be supported by the observation of a main effect of Error Type and an interaction between Error Type and Target Type.

## Results

Error rates for confidence trials are shown in Fig 5. The Bayesian repeated measures ANOVA showed that the full model, with main effects for error type, target type, and the interaction between the two had the best fit (BFm >1000, see S1 Appendix for full model comparison and Table 5 for an overview of BFincl for each model component). Expectation-induced errors occurred more often than other errors (BFincl > 1000 for Error Type) and this effect of Error Type interacted with Target Type (BFincl > 1000): pseudo-letter targets were more likely than real-letter targets to evoke an expectation-induced illusory response (going from 13% illusions

**Table 5. Error rates ANOVA.**

| Effects | P(incl) | P(excl) | P(incl\|data) | P(excl\|data) | BF$_{incl}$ |
|---|---|---|---|---|---|
| Error Type | 0.600 | 0.400 | 1.000 | $3.331^{e-16}$ | $2.002^{e+15}$ |
| Target Type | 0.600 | 0.400 | 1.000 | $7.180^{e-13}$ | $9.285^{e+11}$ |
| Error Type $^*$ Target Type | 0.200 | 0.800 | 1.000 | $8.764^{e-7}$ | $4.564^{e+6}$ |

for real-letter targets to 30% illusions for pseudo-letter targets, BF>1000), while the error rate for other kinds of errors was not influenced by target-type (4% for and real and 5% for pseudo-letter targets, BF = 0.28).

## Discussion

In contrast to Experiment 1–2, the stimulus set used in Experiment 3 had equal numbers of real- and pseudo-letter targets: Each memory display and each response display consisted of equal numbers of pseudo- and real-target letters, and there were equal numbers of trials in which the target was a real and a pseudo-letter. As in Experiment 1–2, the results of Experiment 3 show that illusory memories occur in both real- and pseudo-letter trials, but, also similar to Experiment 1–2, people are more likely to have an expectation-induced illusory memory when the target was a pseudo-letter compared to a real letter.

The tendency to report more illusions in pseudo- than real-letter targets in Experiment 1–2 could have been due to the overrepresentation of real letters targets in the experiment. However, Experiment 3 shows that this is a highly unlikely explanation for the findings in Experiment 1–2: when letters and pseudo-letters are just as prevalent, participants still show the same pattern of results as observed in Experiment 1–3, with a stronger tendency to report, with high confidence, that they have seen a real letter, when in fact they were shown a pseudo-letter.

However, Experiment 3 did not test whether expectancy induced memory illusions also increased over time, as the memory trace for the memory display declined. In Experiment 4, we therefore repeated Experiment 3, but now included a short and a long memory delay, as in Experiments 1–2.

## Experiment 4

### Participants

In this experiment, 95 participants participated (28 men; age range 17–41 years, average age 20.8 years), all having normal or corrected-to-normal vision. All participants were naïve to the purpose of this experiment. All participants gave their informed consent online to participate in the study, which was approved by the local ethics review board of the University of Amsterdam. The participants participated for student credits within the first-year Psychology curriculum. Sample size was determined based on a stopping rule implemented in an ongoing sequential analysis of the Bayes Factor for the critical t-test, namely the comparison between the effect of Error-Type for Real and Pseudo-letter targets. Every other day of testing, the sequential analysis was run, and once the outcome crossed a BF of 10, testing would stop.

### Stimuli

The memory displays and irrelevant intervening screens were the same as in Experiment 3.

### Procedure

The experiment consisted of 18 blocks of 25 trials. The procedure was identical to Experiment 3, with 2 exceptions. First, the interval between the memory display and the cue was either 500 ms (short interval) or 3000 ms (long interval). Second the data was collected in an online experiment, using Neurotask scripting [31]. Participants that signed up to take part in the experiment were invited to join a Zoom meeting at a preset time. Here, each participant was placed in a Zoom break-out room, and given the link to the Neurotask experiment, which they then started on their own laptop. Via the share screen option, participants were given instructions and feedback during the practice block, which otherwise proceeded identically to

Experiment 3. After the practice block, the experiment leader left the breakout room, but was still in the main Zoom session, and available for questions via chat until the participant finished the experiment.

## Analysis

The relative occurrence of illusory memories (on high confidence trials) was analyzed in an ANOVA with target type (real or pseudo letter), interval (short or long) and type of error (expectation-induced illusory error—from pseudo-to-real or vice versa—or other error) as independent variables.

Exclusion of participants with too few high confidence trials in one or more cells of the design (10 trials or less) resulted in an inclusion of 91 participants (4 participants were excluded). Included participants had on average 50.6% high confidence trial over all conditions (range: 13,8–86.4%, sd: 15.7%). Excluded participants had on average 8.5% high confidence trials over all conditions (range 0.4–10.7%, sd: 2.6%). Hypothesis 1 (Illusory memories occur in STM), Hypothesis 2 (These illusions are memory-driven) and Hypothesis 3 (The illusory memories are driven by expectations about the world) are tested, using a stimulus set that contains the same amount of world knowledge confirming stimuli (real letters) as novel stimuli (pseudo-letters). Hypothesis 1 would be supported if expectation-induced errors are more prevalent than other errors, resulting in a main effect of Error Type. Hypothesis 2 would be supported if high confidence illusory reports become more prevalent with longer memory delays, resulting in an interaction between Error Type and Memory Delay. Hypothesis 3 would be supported if the high confidence illusory reports (but not other errors) are more likely to occur when the target in the memory display is a pseudo-letter compared to a real letter, resulting in an interaction with Target Type and Error Type.

## Results

Fig 6 shows the error rates for high confidence trials. A Bayesian repeated measures ANOVA model comparison showed that the model with main effects for Error Type, Target Type, Memory Delay, and all 2-way interactions (but not the 3-way interaction), had the best fit (BFm >1000, see S1 Appendix for full model comparison and Table 6 for an overview of BFincl for each model component). Participants were more likely to select the mirrored

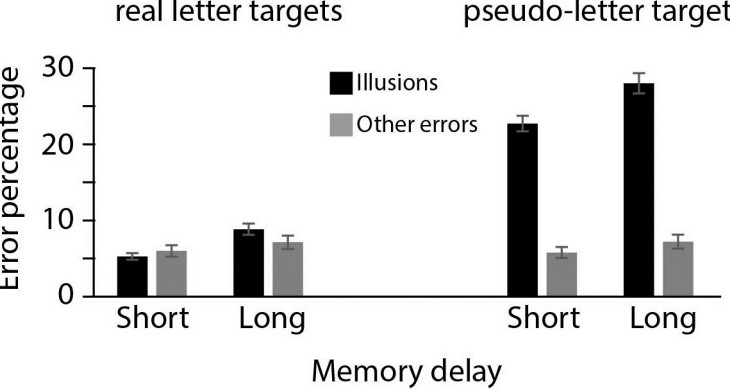

**Fig 6. Experiment 4.** Error rates in trials where participants indicated high confidence in their response. Percentages of expectation-induced errors and other errors are depicted for pseudo-letter targets (right display) or a real letter targets (left display). All error rates are reported for the short and long Memory Delay conditions.

**Table 6. Error rates ANOVA.**

| Effects | P(incl) | P(excl) | P(incl\|data) | P(excl\|data) | BF$_{incl}$ |
|---|---|---|---|---|---|
| Target Type | 0.737 | 0.263 | 1.000 | 1.81$^{e-14}$ | 1.97$^{e+13}$ |
| Memory Delay | 0.737 | 0.263 | 1.000 | 1.91$^{e-6}$ | 186549.94 |
| Error Type | 0.737 | 0.263 | 1.000 | 1.81$^{e-14}$ | 1.97$^{e+13}$ |
| Target Type * Memory Delay | 0.316 | 0.684 | 0.166 | 0.83 | 0.431 |
| Target Type * Error Type | 0.316 | 0.684 | 1.000 | 1.81$^{e-14}$ | 1.20$^{e+14}$ |
| Memory Delay * Error Type | 0.316 | 0.684 | 0.843 | 0.16 | 11.66 |
| Target Type * Memory Delay * Error Type | 0.053 | 0.947 | 0.024 | 0.98 | 0.45 |

counterpart of a target than to make other types of errors. This effect of error-type (Bfincl > 1000) interacted with the type of target (Bfincl>1000). Pseudo-letter targets were more likely than real-letter targets to evoke an illusory erroneous response (going from 7% illusions for real-letter targets to 26% illusions for pseudo-letter targets: BF>1000 in a follow-up t-test), while the other (non-illusory) error rate was not influenced by target-type (7% for real- and pseudo-letter targets: BF = 0.12 in a follow-up t-test).

The tendency to erroneously select the mirrored counterpart of the target increased with a longer memory delay, more so than to make other (non-illusory) errors, resulting in an interaction between Memory delay and Error type (Bfincl = 11.66). Illusory responses increased from 14% to 19% in the longer delays (BF > 1000 in follow-up t-test), while the other non-illusory error rate only increased from 6% to 7% (BF = 4.84 in a follow up t-test).

There was a smaller increase in illusory errors from short to long delays when the target was mirrored (5% difference) compared to when the target was a real letter (4% difference). For other errors, the difference in the effect of memory delay between the two target types was the same (1% increase from short to long delays for real letter targets and for mirrored targets). However, the best-fitting model did not include the three-way interaction between Memory Delay, Error Type and Target Type, with a Bfincl = 0.45 showing anecdotal evidence against including the three-way interaction).

## General discussion

Four experiments consistently show that illusory memories can arise even when the to-be-remembered items have only just disappeared from vision. Participants consistently report, with high confidence, that they have seen the real counterpart of a pseudo-letter target. These memory illusions seem to be the result of world knowledge (i.e., based on the usual orientation of letters), and not of visual similarities, as they are much more prevalent than real-to-pseudo illusions.

Illusory memories make up almost 20% of reports when memory is probed at 500 ms (Experiment 1 and 2) and up to 30% of reports when memory is probed at 3 seconds (Experiment 1–4). Moreover, Experiments 3 and 4 show that pseudo-to-real letter illusory memories are also more prevalent than real-to-pseudo illusory memories when the stimulus set is completely balanced so that pseudo- and real letters occur equally often in the memory displays and as targets. Taken together, the results thus show that world knowledge can shape memory even when memories have only just been formed.

Experiments 1, 2 and 4 show that illusory memories make up a larger part of high confidence responses as time passes (from 0.5 s to 3 s). This suggests that as the quality of the memory representation decreases through decay or interference, world knowledge plays an increasingly large role within STM. Also apparent from this increase in illusory memories over

time, is that these incorrect responses are not just the result of errors in initial perception of the visual stimulus, or in encoding the stimulus incorrectly into memory. One can argue that expectations could very well shape perception, leading people to *see* (and thus report) a true letter when the visual input was a pseudo-letter. However, the increase in memory illusions over time suggests that, in our experiments, participants mainly start out with a correct percept, which they quite accurately report when probed early after the memory display has disappeared. Over time illusions become more prevalent, indicating that correctly encoded percepts of a pseudo-letter are being converted to illusory memories of a real letter. So, perceptual illusions can, at best, account for a small number of incorrect responses, but memory illusions are also clearly present.

### The origin of illusory memories in STM

How can world knowledge shape the memory of events that have only just been removed from vision? One possibility is that both perception and cognitive processes such as STM operate according to Bayesian principles. According to these principles, as expressed in frameworks such as 'predictive processing' [32–35], top-down prior expectations are integrated with bottom-up input to form perceptual or cognitive content. This mechanism allows that world knowledge can play a significant role within perception, as has now been extensively demonstrated [36–38]. The present study shows that similar effects also apply to the formation and recall of visual short term memory (STM).

In predictive processing accounts of perception and cognition, the role that top-down information plays within cognition depends on the relative strength (or precision) of the bottom-up and top-down signals. In our experiments, top-down expectations for real letters can be expected to be relatively strong, as participants have been exposed to the standard alphabet frequently and from a young age. Pseudo-letters, on the other hand, will not have strong internal representations. Therefore, our finding that pseudo-letters are much more likely to give rise to real letter memory illusions, than the other way around, is expected: Memory traces for pseudo-letters are combined with strong internal priors for the real counterparts, making them more likely to result in illusory memories for the real letter. For real letter-targets the bottom-up memory information matches the internal predictions, and therefore the chance of formation of an illusory memory for the pseudo-letter counterpart of the real letter is small (although participants still sometimes select the pseudo-version of a real letter, but the incidence of these errors approximates the chance of selecting any other item in the display).

One result that is particularly in line with a generative processing view of memory, is the finding that illusory memories become more prevalent as the memory representation decreases in strength, whether over time and/or through interference—reflecting a decrease in the strength of the relevant 'bottom up' signal [38]. Indeed, in our experiments, when the bottom-up memory trace from the original memory display weakens, because of decay or interference, the internal predictions increasingly dominate the high confidence memory reports of the participants.

An alternative to the idea that STM illusions are based on predictive processing, could be that the strong priors strengthen the encoding of the memory representation and make it less susceptible to decay. Indeed well-known real-life categories are linked to a very strong, well established representation in memory while lesser known stimuli like pseudo-letters are not [39–45]. This would make it easier to differentiate the memory trace for a real letter target compared to a pseudo-letter target [46] making the memory representation of the pseudo-letter weaker, and more prone to errors, and thus illusions. Moreover, it will also make it more likely that the representation of the pseudo-letter weakens (through decay or interference)

compared to the strongly differentiated real letter representations, thus making pseudo-letters more prone to errors and illusory memories over time and interference. However, our findings clearly show that pseudo-letter targets do not simply lead to weaker memory traces. If that were the case, participants remembering a pseudo-letter target (or more accurately, a participant easily forgetting a weakly established representation of a pseudo-letter target) should be selecting all items in the probe display with equal probability, as the representation of the target has been lost. This should result in similar rates of errors (selecting probe items that are not the target, or a mirrored version of the target) and illusions (selecting the mirrored version of the target). Our results show that this is not the case: Illusory memories are much more common than other errors when the target was a pseudo-letter. Moreover, the incidence of expectation induced illusory memories increases as the memory trace weakens, and this is not (or rather, much less) the case for other errors.

Although predictive processing provides a comfortable framework with which to account for our results regarding STM, we recognize that there is already a vast literature about the theoretical foundations of false memory in LTM. For example, one explanation for the occurrence of illusions in LTM comes from fuzzy trace theory [47]. FTT states that memory consists of two parts, a verbatim part, representing the exact memory input, and a gist part, representing the meaning of the input, based on a semantic analysis. Could FTT also explain the current findings in STM? It is possible that the verbatim representation of the visual input is the physical pseudo-letter, and that the high-level gist that is stored is something like "letter" and "mirrored". With such a representation, it is possible that the "letter" gist is stronger than the "mirrored" gist, thus surviving longer. However, FTT predicts that high similarity between a target and its mirrored counterpart should inhibit false memories [48]. This is not consistent with the current data: The mirrored counterpart (the real letter) and the target (the pseudo-letter) are extremely similar, yet false memories are abundant. Therefore, it seems that FTT cannot accurately account for the occurrence of this type of memory illusions in STM. In contrast to FTT, which suggests that memory illusions are the result of high-level semantic analysis, it thus seems that, at least for memory illusions in STM, very basic perceptual expectations can directly shape memory representations. Therefore, based on these considerations we argue that FTT cannot entirely explain the current findings.

FTT does suggest that pseudo-letters have a verbal (the letter itself) and a gist level ("mirrored") representation, while real letters can be stored purely at the verbal level. This would mean that verbal rehearsal is more effective to store a complete representation of real letters in working memory, but not for pseudo-letters. Removing verbal rehearsal as a strategy, for example through articulatory suppression, would therefore impact the storage of the verbal representation in working memory, leading to a decayed memory trace for real letters and pseudo-letters. It is worthwhile to test, in future research, whether more decayed memory traces for real letters are similarly sensitive to memory illusions (real-to-pseudo) as pseudo-letter representations are in our experiments. This would clarify whether the current effects are the result of maintenance effects (real letters are more effectively maintained than pseudo-letters and therefore less sensitive to top-down modulation) or prediction-related effects at retrieval (strong internal priors are more likely to replace ambiguous memory representations).

## The pervasiveness of illusory memories in STM

STM is often subdivided into Iconic Memory and Working memory. Iconic memory (IM), is a short-lived (<500 ms) high capacity memory storage [1], while working memory longer lasting, but with a smaller capacity of 2 to 4 items [2]. It has been suggested that Fragile Memory

constitutes an intermediate form of STM [3], which, like WM, lasts at least 4 seconds, but unlike WM has a capacity of 5–15 items and is destroyed by visual interference [24]. Our findings suggest that illusory memories can occur during the WM and FM stages of STM. Experiment 1 shows that when the memory cue is presented together with unrelated information, a trial-type most like a WM manipulation, illusory memories are highly prevalent. Experiment 2–4 show that in a FM set-up illusory memories are also abundant. Our experiments currently do not test whether illusory memories arise in IM, which is sometimes thought not to be susceptible to illusions based on contextual information or internally generated biases [20].

## The implications of STM illusory memories for vision research

The fact that all stages of STM are susceptible to illusory memories has important implications for vision research, as WM and IM tasks are often used to investigate the content of the visual percept for example in visual masking tasks [49, 50] or in change blindness tasks [51, 52]. The current experiments show that internal expectations can quite dramatically change the memory of the perceiver. Therefore, if a perceiver indicates having seen something that is different from the input, it may be difficult to rule out that this change is a result of a memory illusion, instead of a perceptual illusion. As mentioned before, De Gardelle and colleagues (De Gardelle et al., 2009) [7] showed memory displays which contained rotated letters. In a WM task, participants indicated seeing the real counterpart of the rotated letter. De Gardelle and colleagues (2009) [7] conclude that this indicates a perceptual illusion: They claim the perceivers *saw* something different than the actual bottom-up input because of their internal expectations. However, the current research shows that this effect could also be due to a memory illusion: Perceivers might have seen the actual pseudo-letter, but just moments later *remembered* seeing the real letter. Moreover, Experiments 1, 2 and 4 show that this is perhaps a more likely explanation than an illusion that occurs when perceiving or encoding the stimuli: In these 3 experiments, expectation-induced illusions became more prevalent as the delay between seeing the memory display and the memory probe increased. This shows that the illusions come into being while the memory trace is already (correctly) stored in STM. Intriguingly, research employing an IM task, suggests that illusory memories for letters can arise even when no stimuli were present at all [53, 54], which suggests that even when no sensory memory trace is originally stored in memory, strong expectations can nonetheless give rise to memory illusions.

Chalk and colleagues [55] showed that participants were biased to report faintly visible dots moving in one specific direction after having been consistently exposed to that particular direction of motion for multiple trials. In reality, the dots on these test trials were not moving in the direction of the previous trials. Again, the authors interpreted this as a perceptual bias, but since perception was only probed after the visual stimulus display had disappeared, this could just as well be a memory bias. If these findings are indeed the result of a memory bias, the Chalk et al. study (2010) has two very interesting implications. First of all, it would show that illusory memories can arise from external information as well as internal priors. Secondly, it would suggest that memory biases are not limited to letters, but instead apply to (visual) percepts in general. This implicates that very early illusory memories could be an inherent feature of the cognitive processing sequence for sensory input.

In conclusion, four experiments consistently show that illusory memories arise even when the visual stimulus has only been out of sight for very brief periods of time. These results show that even the most recent recollections are susceptible to illusory memories. Moreover, it suggests that internal priors play a crucial role not just during perception, but also in memory. Thus, it seems that short-term memory is not always an accurate representation of what was

just perceived. Instead, memory is shaped by what we expected to see, right from the formation of the first memory trace.

## Supporting information

**S1 Appendix.**
(DOCX)

## Author Contributions

**Conceptualization:** Marte Otten, Anil K. Seth, Yair Pinto.

**Data curation:** Marte Otten, Yair Pinto.

**Formal analysis:** Marte Otten, Yair Pinto.

**Investigation:** Marte Otten, Yair Pinto.

**Methodology:** Marte Otten, Anil K. Seth, Yair Pinto.

**Project administration:** Marte Otten, Yair Pinto.

**Validation:** Marte Otten, Anil K. Seth, Yair Pinto.

**Visualization:** Marte Otten, Yair Pinto.

**Writing – original draft:** Marte Otten, Yair Pinto.

**Writing – review & editing:** Marte Otten, Anil K. Seth, Yair Pinto.

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
