## [Decision Letter · Decision Letter 0]

3 Nov 2022

PONE-D-22-23654Seeing Ɔ, remembering C: Illusions in short-term memoryPLOS ONE

Dear Dr. Otten,

Thank you for submitting your manuscript to PLOS ONE. After careful consideration, we feel that it has merit but does not fully meet PLOS ONE’s publication criteria as it currently stands. Therefore, we invite you to submit a revised version of the manuscript that addresses the points raised during the review process.

We look forward to receiving your revised manuscript.

Kind regards,

David K Sewell

Academic Editor

PLOS ONE

Journal Requirements:

Additional Editor Comments:

Dear Dr Otten,

Thank you for submitting your manuscript to PLOS ONE. I have now received reviews from two experts and have also read through the manuscript myself. Reviewer 1 identifies himself as William Ngiam. You’ll see that the reviewers are quite positive about the work—as am I—but that there are some issues that require further discussion and clarification.

I am therefore rejecting this version of the manuscript, but invite you to submit a revision to address the comments and concerns voiced by the reviewers. I highlight points that require particular attention here, but please respond to all points raised by the reviewers.

I am hopeful to be able to make a final decision on a revised version of the manuscript without having to send this out for further review.

Both of the reviewers expressed reservations about the use of “illusion” to refer to the pattern of memory errors your observed. I am sympathetic to these concerns, as the term clearly carries different connotations to different researchers. Providing a more explicit definition of your usage of the term would help. Alternatively, you might consider adopting different terminology altogether. I leave it to your discretion as to how you approach this.

The reviewers are also consistent in their concerns about the overall proportion of high confidence responses upon which the key results depend. Given the relatively small number of trials that feed into your analyses, knowing the rate of high-confidence responses is critical for drawing solid inferences from the data. Related to this, I had concerns over the distribution of high-confidence responses made across participants (see also Reviewer 1’s more specific concerns on this matter). Restricting the analysis to participants who produced high-confidence responses at a sufficient rate would help to ensure the results are not driven by a subset of participants who used the confidence scale more conservatively than others. In a similar vein, are the results/conclusions altered by splitting responses into high confidence (3 and 4) vs. low confidence (1 and 2)?

Reviewer 2 raises questions about the high variability of methodological details and sample sizes across experiments. I too was puzzled by this, as there was no explicit rationale given for this (see also Reviewer 2’s questions about the motivation for the short interference manipulation in Experiment 2). If there were power-related considerations that informed the sample sizes, these should be discussed. Similarly, the motivations for methodological variations should be made explicit. I was curious if the logic ran along the lines of multiverse analyses where aspects of a study methodology are varied randomly to demonstrate robustness of key effects against “nuisance” method variance.

Reviewer 2 also raises an excellent question about the potential role of rehearsal, given the verbal nature of the stimuli and how rehearsal was not controlled for (e.g., via articulatory suppression). I do not think an additional experiment is required here, but some discussion of this potential caveat is important.

Finally, I shared Reviewer 1’s confusion about the way Bayesian statistical analyses were used. While it is true that one of the attractive features of the Bayesian approach is that it allows quantification of support for the null, this is equally true for the alternative hypothesis. Indeed, this goes beyond what traditional NHST approaches can achieve, so it was strange to see the Bayesian analyses incorporated in such a narrow fashion. Given that you’re already adopting Bayesian analyses, it would be sensible to present a fully Bayesian analysis (or include the full set of Bayes factors alongside your significant effects as well).

I thank the reviewers for their constructive comments on this manuscript. I very much hope that you find them similarly useful. I look forward to receiving your revised submission.

Yours sincerely,

David K Sewell

Reviewers' comments:

Reviewer's Responses to Questions

**Comments to the Author**

1. Is the manuscript technically sound, and do the data support the conclusions?

Reviewer #1: Partly

Reviewer #2: Yes

2. Has the statistical analysis been performed appropriately and rigorously? 

Reviewer #1: I Don't Know

Reviewer #2: Yes

3. Have the authors made all data underlying the findings in their manuscript fully available?

Reviewer #1: No

Reviewer #2: Yes

4. Is the manuscript presented in an intelligible fashion and written in standard English?

Reviewer #1: Yes

Reviewer #2: Yes

5. Review Comments to the Author

Reviewer #1: PONE-D-22-23654 Review

SUMMARY

The authors examine whether short-term memories (those formed within a few seconds) can be influenced by top-down expectations, reporting an empirical phenomenon that they refer to as short-term memory illusions. In five experiments, participants were required to do a visual short-term memory task with mixed arrays of real letters or ‘pseudo-letters’ (inverted or mirrored real letters). These arrays were briefly presented (250ms) and then one location was tested on each trial – sometimes a real letter target, otherwise a pseudo-letter target. The response screen contained many choices (sometimes including a ‘none of the above’ choice), amongst which was sometimes the target from the array, or the transformed target. Looking at only high-confidence responses (rating of ‘4’ on a scale from 1 to 4), the authors find that when a transformed letter is probed, participants moreso err by accurately reporting the identity of the letter without its transformation, but when the real letter is probed, participants do not often make this error. The authors characterize this effect as a short-term memory illusion, and conclude that internal expectations play a substantial role in memory as it would in perception.

The authors have conducted a wide range of experiments that attempt to titrate this illusory effect but these experiments appear to vary in many critical ways (set size of memory array, proportion of real/transformed letters in the memory array, intervening arrays, delay interval to test, fixation cross cueing potential target, number of possible choices), making it difficult to follow the guiding theoretical principles and evaluate what about the empirical effect has reliably been established. Despite this, this effect appears to appear fairly consistently across all experiments which leads me to believe it is likely a reliable effect. Although I am favorable to the idea that top-down expectations can influence short-term memories (in many ways) and find this empirical effect particularly interesting, I am not certain whether this effect is best characterized as being a short-term memory illusion. To me, an illusion invokes the idea that the short-term memory representation has frequently and substantially been truly altered, but that has not been clearly demonstrated yet in this manuscript.

Here in my review, I hope to have made comments that lead to better clarity of the research, and better connect the research to the wider visual working memory literature – with the aim of providing useful suggestions to the authors in improving their manuscript. I admit that I could not devote as much effort to conduct a thorough review as I would have hoped, and apologize as a result if this review causes any unintended frustrations.

MAJOR COMMENTS

1. I would like to compliment the authors for committing to sharing these materials on the Open Science Framework, which will better archive the scientific outputs and promote the transparency and reproducibility of the research. However, I wished I could have informed this present review with access to the experimental code and data, and analysis code. One way I would have informed my review with access to open data would be to better understand the proportion of high-confidence responses being made as they vary across the many conditions, and attempt to reproduce the main statistical analyses – I was missing clear reporting on the proportion and variation of high-confidence responses made by each participant, which I think is critical for interpreting these results.

Taking Experiment 1 for example, participants completed 240 trials in total. As there was only one pseudo-letter in the memory array for these experiments, there were only 40 trials where these were tested, and participants would make a high-confidence response on a smaller proportion of that. Further, there were three delay intervals employed (150, 350 and 750 ms) further partitioning the number of trials. Thus, I am concerned about whether any tested effect is a reliable and consistent empirical phenomenon and whether this effect is unduly influenced by different bias across participants to respond with overall high-confidence on the 4-point scale. One initial worry is that perhaps individuals that selectively report high-confidence responses will generally make a substantial contribution to the reported effects (as proportions of errors will be overall larger values). Thus, the illusory effect in the manuscript may be mostly determined by an overall overconfidence in memories rather than specific confusions regarding the memory that is driving their response.

I wanted to note that this effect does not seem contingent on high-confidence responses – confusions of pseudo-letters for real letters still occurred for low-confidence responses albeit at a lower rate (35% compared to 22%, as reported in Experiment 2, p.27). I do appreciate that the confusion rate is higher with high-confidence responses, a perhaps surprising trend, but with a bit of uncertainty around these estimates, I’m uncertain on how to best draw an inference. I think it would be worthwhile to present in the manuscript how these responses change as a function of confidence across all experiments to show that there is an illusory effect that happens moreso with high-confidence responses.

2. The increase in memory illusions over time (from short delay to long delay) was interpreted to mean that “correctly encoded percepts of a pseudo-letter are being converted to illusory memories of a real letter” (p. 46). One could suggest that the reason that this effect is not bi-directional (real letters being converted to pseudo-letters) is because familiar, real letters are better maintained, resistant to interference or confusion. There is evidence that familiarity, or real-word stimuli such as letters (apologies for the self-citation), are afforded better processing into and consolidation within working memory:

Ngiam, W. X., Khaw, K. L., Holcombe, A. O., & Goodbourn, P. T. (2019). Visual working memory for letters varies with familiarity but not complexity. Journal of Experimental Psychology: Learning, Memory, and Cognition, 45(10), 1761.

Brady, T. F., Störmer, V. S., & Alvarez, G. A. (2016). Working memory is not fixed-capacity: More active storage capacity for real-world objects than for simple stimuli. Proceedings of the National Academy of Sciences, 113(27), 7459-7464.

Jackson, M. C., & Raymond, J. E. (2008). Familiarity enhances visual working memory for faces. Journal of Experimental Psychology: Human Perception and Performance, 34(3), 556.

Xie, W., & Zhang, W. (2017). Familiarity increases the number of remembered Pokémon in visual short-term memory. Memory & cognition, 45(4), 677-689.

Asp, I. E., Störmer, V. S., & Brady, T. F. (2021). Greater visual working memory capacity for visually matched stimuli when they are perceived as meaningful. Journal of Cognitive Neuroscience, 33(5), 902-918.

Hedayati, S., O’Donnell, R. E., & Wyble, B. (2022). A model of working memory for latent representations. Nature Human Behaviour, 6(5), 709-719.

Brady, T. F., & Störmer, V. S. (2022). The role of meaning in visual working memory: Real-world objects, but not simple features, benefit from deeper processing. Journal of Experimental Psychology: Learning, Memory, and Cognition, 48(7), 942.

I appreciate the framework that the authors describe in the ‘The origin of illusory memories in STM’ section in the discussion (p. 47-50) – that top-down effects can be produced by predictive processing or protection from decay. To me, this entails differences in either encoding into short-term memory stores or consolidation within the store. I wonder if the authors could better specify their theory or the mechanism by which the illusion is happening here, as to better clarify their results.

Therefore, I am unsure whether the authors can make a strong case for illusory memories here, when the effect may be best described as a function of both guessing and familiarity benefits. One experiment design that I think would be informative in disentangling real-world experience of stimuli and current situational beliefs about the stimuli (say, the proportion of real-world letters and transformed letters in the arrays) would be reversing the proportion of transformed letters to real-world letters (more transformed letters than real letters) – this should manipulate the participant’s expectations about whether the target is more likely a pseudo-letter and you might see a ‘reversal’ of the illusory effect shown here, and a better specification of the effect.

3. In Experiment 3, the color of the fixation cross acted as a cue – red indicated the pseudo-letter was the target on 25% of trials, blue indicated the pseudo-letter was a target on 60% of trials. The authors report the participants were not consciously aware of this pattern, before collapsing across these conditions in subsequent analyses. Could the authors indicate how this was tested, such as whether the participants were explicitly tested on these percentages, or whether they noticed any changes in likelihood of the pseudo-letter being a target?

In addition, was there any error correction applied for any post-hoc comparisons? For example, such as with the pairwise comparisons between the various conditions in Experiment 2 (short+interference versus long, short versus long), or comparing within high-confidence trials to within low-confidence trials. If there was any error correction applied, it would be best to explicitly mention that in the manuscript.

MINOR COMMENTS

1. I think the Bayes Factors in the manuscript might need a check through, perhaps using the BF10 or BF01 notation to indicate the direction of the likelihood ratio.

On page 15, there is a typo in reporting the Bayes Factor in text – “However, there [was] no significant interaction between error type and time (F(2,106) = 1.68, p = .19, eta^2 = 0.031, BF = 0.131). The Bayes Factor of .31 indicates substantial evidence for the Null hypothesis.”

On page 45, the second paragraph, the Bayes Factor reported is 2.2, but it is described in-text as ‘only showed anecdotal evidence for the absence of an effect’. If this BF is consistent with how it is reported in the manuscript, this would be evidence in favor of the alternative hypothesis, rather than for the absence of the effect.

I think the Bayes Factor should be reported for all analyses in the manuscript, rather than left to the Appendix. I think the way it is reported presently in the manuscript suggests or encourages that the Bayesian analysis only be employed to evaluate evidence for the null hypothesis. While an advantage of Bayesian analyses is that it can address inferential issues around null hypothesis significance testing by providing likelihood ratios for competing hypotheses, it should not be used to selectively evidence particular results such as the null hypothesis. I do see how reporting all the results may crowd the results paragraph, so I leave this up to the authors’ discretion on how they report the values.

2. On page 42, last paragraph, there is a typo in – “As in Experiment 5, Hypothesis 1…” – I believe the authors meant Experiment 4 here.

3. The term ‘psuedo-letter’ is used throughout the manuscript, but I think I prefer the term ‘transformed’ or in later experiments ‘mirrored letter’. There exists non-letter character sets like the Brussels Artificial Character Sets, which are artificial letters so as to better distinguish non-letter stimuli, letter-like stimuli and the manipulations in the present experiment.

4. The y-axis in all figures are labelled as ‘percentage of trials’, which I found confusing to parse. I initially interpreted this as percentage of total trials in the experiment, rather than a proportion of ‘high-confidence trials’ for example.

5. This effect reminds me of ‘swap errors’ and ‘feature misbinding’ errors that have been documented in the visual working memory literature. I think a potentially fruitful thread would be to consider whether the ‘identity’ of the letters are apprehended faster or better encoded than the ‘transformation’ of the letters – it might be worthwhile considering a mechanism whereby the retrieval of pseudo-letters may be via encoding the transformation of the letter anchored to the upright real letter (and that this transformation is decayed). I’ve attached some references here where there has been debate as to what generates these kinds of errors, whether they are guesses or a function of feature variability.

Bays, P. M. (2016). Evaluating and excluding swap errors in analogue tests of working memory. Scientific reports, 6(1), 1-14.

Fougnie, D., & Alvarez, G. A. (2011). Object features fail independently in visual working memory: Evidence for a probabilistic feature-store model. Journal of vision, 11(12), 3-3.

Pratte, M. S. (2019). Swap errors in spatial working memory are guesses. Psychonomic bulletin & review, 26(3), 958-966.

McMaster, J. M., Tomić, I., Schneegans, S., & Bays, P. M. (2022). Swap errors in visual working memory are fully explained by cue-feature variability. Cognitive psychology, 137, 101493.

William Xiang Quan Ngiam

I sign all my reviews, regardless of the recommendation to the editor. By signing this review, I affirm that I have made my best attempt to be polite and respectful while providing criticism and feedback that is hopefully helpful and reasonable.

Reviewer #2: In the present manuscript, the authors perform five behavioral experiments to evaluate “illusions” in short term memory, and whether they are perceptual effects or if they arise in working memory. Specifically, participants remember 6-8 letters, a subset of which are “pseudo” (mirrored) letters. After a cue and some delay, participants are probed to report the cued letter. Participants are more likely to mistake a pseudo letter for its real counterpart than they are to mistake a real letter for its pseudo letter counterpart, and this effect is more pronounced with increasing delay. As such, the authors conclude that short term memory illusions are not perceptual errors but arise in memory.

I think the premise of this paper is interesting and that the experiments are thorough. I particularly appreciated the inclusion of experiments 4 and 5, as I found myself very much stuck on the imbalanced memory and probe displays in experiments 1-3 and was glad to see those concerns addressed here. I do have some remaining concerns, which I detail below in (mostly) the order that they arose.

1. I did have somewhat of a hard time with the use of the word illusion in this context. As someone who is admittedly more of a vision researcher, to me, the canonical definition of an illusion feels necessarily perceptual (i.e., where visual illusions arise in low-level visual areas). It seems to me that “illusion” is a term used in long-term memory research to be synonymous with false memories, and that in this context it’s meant to reflect false shorter term memories rather than an illusion in the (what I consider) traditional sense. My sense is that I won’t be the only reader with this hang-up, but that it could be quelled early in the manuscript by more clearly introducing the concept of illusions in long term memory and what exactly is meant by illusions in this context.

2. De Gardelle et al (2009) is quite relevant to the current work and should be more clearly detailed when it is first introduced. For example, on my first read through it wasn’t clear to me what the “real counterpart of the rotated letter” meant, and how that was relevant to the central theme of this paper.

3. What are the trial counts in the critical conditions in the experiments? In experiment 1, there are only ~300 trials, and ~100 in each of the delay conditions. It’s not reported how many of these are high confidence trials, but only ~10% of high confidence trials in each condition are meaningful errors, which means there are < 10 trials per cell. Relatedly, the sample sizes change quite dramatically across some of the experiments. It’s particularly large for experiment 3, and the justification for this wasn’t clear. Did the authors run any power analyses to determine sample sizes/trial counts? And what was the stopping rule for data collection?

4. Related to the above, it would be helpful to see the breakdown of high vs. low confidence responses. The critical analyses only include high confidence responses, but it’s not reported what proportion of trials were high confidence.

5. Experiment 2: It’s not clear why the proportions of target/response options was changed here. If the target is a pseudo-letter, its counterpart (a memory lure) is presented as a response option 50% of the time, but if the target is a real letter, it’s pseudo version (memory lure) only appears as a response option 25% of the time. Doesn’t that make them more likely to make a pseudo to real error on a higher proportion of trials than a counterpart to pseudo error? After seeing experiments 4 and 5, I’m mostly convinced that these regularities aren’t sufficient to cause the critical patterns, but I think the choice to change these proportions in experiment 2 should be more clearly motivated (why was this change made?).

6. The logic behind including the short with interference condition is a bit unclear. Was this just to make the task more difficult/what was the prediction?

7. I wonder about the contributions of verbal vs. visual working memory to overall performance in this task and, more importantly, to the systematic bias in response errors. Given that the authors are interested in comparing perceptual vs. memory-based errors, it makes sense to use a task that is designed to rely on visual working memory. With stimuli like letters, however, it’s likely that participants are using the cue to rehearse the target letter using verbal working memory. Verbalizing the target in this way (i.e., rehearsing ‘B’ without rehearsing its pseudo vs. real letter classification) may contribute to the higher instance of pseudo letter to real counterpart illusions, particularly over time as memory starts to decay and reliance on verbal working memory increases. I appreciate that the data focus on high confidence errors, but I think this type of rehearsal lends itself to such errors. Is it possible that the specific pattern of errors reported here arise from a reliance on verbal working memory? If so, does that affect the interpretation of the data, or might it simply be the case that the mechanism underlying these illusions is increased reliance on more course information in memory, such as category information rather than a precise visual representation? I would be curious to see the experiment done with a concurrent articulatory suppression task. I don’t think this is necessarily required, but I am curious to hear the authors’ thoughts on this, either in a response or in the manuscript.

Minor:

- Figure 3: It wasn’t immediately clear to me the importance of the dotted line at 20% in the left panel. Also, for this and subsequent similar graphs, you might consider changing the label on the x axis to Time from array offset (rather than time to offset, which I found a little confusing).

- Page 18 mistakenly says “occurred either appear”

- Page 15 – there was no significant interaction between error type and time (missing word “was”)

- Regarding the way timing is indicated on the trial schematics: rather than 0.75 – 3 s, it might be clearer to label them.75 or 3 s

- Was there really a 1.5 to 3.75 s delay following the probe on trials where the probe and irrelevant letter display were sequential as well? In-text suggests that the second display onset 5 s after the offset of the first letter display, but wouldn’t this depend on whether the trial belonged to the short or long delay condition? If this labeling is correct, then in the 3 s condition, the delay between offset of the first letter display and onset of the second could be over 7 seconds.

- Some of the language regarding support for the various hypotheses is a bit strong throughout. Specifically, language that describes the hypotheses as “confirmed” rather than, for example, supported.

- Were the letters in the first and second letter displays (the memory array and the interference display) selected randomly with or without replacement? (Could letters reappear in the interference display?)

- The intro says there are four experiments but the paper includes 5

6. PLOS authors have the option to publish the peer review history of their article (what does this mean?). If published, this will include your full peer review and any attached files.

Reviewer #1: **Yes: **William Xiang Quan Ngiam

Reviewer #2: No

---

## [Author Response · Author response to Decision Letter 0]

18 Jan 2023

Below, we address the comments from the editor and the two reviewers. Before doing that, we want to thank the editor and the two reviewers for their helpful and thoughtful comments.

Editor comments

Editor Comment 1

Both of the reviewers expressed reservations about the use of “illusion” to refer to the pattern of memory errors your observed. I am sympathetic to these concerns, as the term clearly carries different connotations to different researchers. Providing a more explicit definition of your usage of the term would help. Alternatively, you might consider adopting different terminology altogether. I leave it to your discretion as to how you approach this.

Reply

To clarify our choice for the term illusion, we have added the following section to page 3:

“These experiments specifically focus on a subset of false memories, namely what we call “memory illusions”: cases were people report with high confidence to have seen something that was not presented to them, i.e., the illusory experience of a reliable and reportable memory. An example of this would be clearly remembering that you put your keys on the diningroom table, while in reality you left them on the side table by the front door. Focusing on memory illusions instead of general false memory reports allows a separation of such incorrect but seemingly true memories from memory lapses that are filled in by best guesses (such as when you have no clue where you put your keys, but you check out the dining room table, because you very often drop them there). In such memory lapses, the reliance on pre-existing knowledge is a useful strategy to consciously fill gaps in memory, but it does not reveal whether prior knowledge can actually shape the content of STM.”

Editor Comment 2

The reviewers are also consistent in their concerns about the overall proportion of high confidence responses upon which the key results depend. Given the relatively small number of trials that feed into your analyses, knowing the rate of high-confidence responses is critical for drawing solid inferences from the data. Related to this, I had concerns over the distribution of high-confidence responses made across participants (see also Reviewer 1’s more specific concerns on this matter). Restricting the analysis to participants who produced high-confidence responses at a sufficient rate would help to ensure the results are not driven by a subset of participants who used the confidence scale more conservatively than others. In a similar vein, are the results/conclusions altered by splitting responses into high confidence (3 and 4) vs. low confidence (1 and 2)?

Reply

We have adapted the analyses to only include participants who had at least 10 high confidence trials in each of the relevant cells of the design. This meant, unfortunately, that we had to remove experiment 1 completely, as there were only 40 trials in total in that experiment that were potentially relevant, and the number of high confidence trials per participant per cell in the design was always below 10. For the remaining 4 experiments, we now report the results only for participants that had more than 10 high confidence trials per condition. For each experiment, we report the overall mean percentage of high confidence trials (plus range and sd) for included and excluded participants. 

Editor Comment 3

Reviewer 2 raises questions about the high variability of methodological details and sample sizes across experiments. I too was puzzled by this, as there was no explicit rationale given for this (see also Reviewer 2’s questions about the motivation for the short interference manipulation in Experiment 2). If there were power-related considerations that informed the sample sizes, these should be discussed. Similarly, the motivations for methodological variations should be made explicit. I was curious if the logic ran along the lines of multiverse analyses where aspects of a study methodology are varied randomly to demonstrate robustness of key effects against “nuisance” method variance.

Reply

We like the concept of the nuisance method, and wish it were an explanation for the variations between the experiments in this manuscript, but reality is more mundane than this: this is a collection of studies that were run over a long period of time as MA thesis projects. Design choices were driven by additional interests of the student, in addition to the main question of honing in on the illusory memory effect. 

With regards to variation in sample size:

Experiment 1 and 2 consisted of a convenience sample.

The very large sample of experiment 2 was part of a big research credit programme in which all first years participated, so the number of participants was not controlled by us. We make this explicit in the methods section of Experiment 2:

“The participants participated for student credits as part of a course within the first-year Psychology curriculum. The experiment was part of a 2-hour session which contained this experiment, and several other unrelated experiments. The number of participants was determined by student enrollment in the course.”

In experiment 3 and 4 we applied a Bayesian stopping criterium. We have included a description of this stopping criterion in the methods section of experiment 3 and refer to it in Experiment 4. 

“Note that in this experiment the number of participants was determined based on a stopping rule implemented in an ongoing sequential analysis of the Bayes Factor for the critical t-test, namely the comparison between the effect of Error-Type for Real and Pseudo-letter targets. Every other day of testing, the sequential analysis was run, and once the outcome crossed a BF of 10, testing would stop.”

Editor comment 4

Reviewer 2 also raises an excellent question about the potential role of rehearsal, given the verbal nature of the stimuli and how rehearsal was not controlled for (e.g., via articulatory suppression). I do not think an additional experiment is required here, but some discussion of this potential caveat is important.

Reply 

We have added this suggestion to the general discussion on page 40

“FTT does suggest that pseudo-letters have a verbal (the letter itself) and a gist level (“mirrored”) representation, while real letters can be stored purely at the verbal level. This would mean that verbal rehearsal is more effective to store a complete representation of real letters in working memory, but not for pseudo-letters. Removing verbal rehearsal as a strategy, for example through articulatory suppression, would therefore impact the storage of the verbal representation in working memory, leading to less stable memory representation of real letters. It is worthwhile to test, in future research, whether memory traces for real letters which cannot rely on verbal rehearsal are similarly sensitive to memory illusions (real-to-pseudo) as pseudo-letter representations are in our experiments. This would clarify whether the current effects are the result of maintenance effects (real letters are more effectively maintained than pseudo-letters) or prediction-related effects at retrieval (strong internal priors are more likely to influence ambiguous memory representations).”

Editor Comment 5

Finally, I shared Reviewer 1’s confusion about the way Bayesian statistical analyses were used. While it is true that one of the attractive features of the Bayesian approach is that it allows quantification of support for the null, this is equally true for the alternative hypothesis. Indeed, this goes beyond what traditional NHST approaches can achieve, so it was strange to see the Bayesian analyses incorporated in such a narrow fashion. Given that you’re already adopting Bayesian analyses, it would be sensible to present a fully Bayesian analysis (or include the full set of Bayes factors alongside your significant effects as well).

Reply

Thanks for this suggestion. We have replaced the NHST analyses with full Bayesian analyses throughout the paper.

Reviewer 1 comments 

The authors examine whether short-term memories (those formed within a few seconds) can be influenced by top-down expectations, reporting an empirical phenomenon that they refer to as short-term memory illusions. In five experiments, participants were required to do a visual short-term memory task with mixed arrays of real letters or ‘pseudo-letters’ (inverted or mirrored real letters). These arrays were briefly presented (250ms) and then one location was tested on each trial – sometimes a real letter target, otherwise a pseudo-letter target. The response screen contained many choices (sometimes including a ‘none of the above’ choice), amongst which was sometimes the target from the array, or the transformed target. Looking at only high-confidence responses (rating of ‘4’ on a scale from 1 to 4), the authors find that when a transformed letter is probed, participants moreso err by accurately reporting the identity of the letter without its transformation, but when the real letter is probed, participants do not often make this error. The authors characterize this effect as a short-term memory illusion, and conclude that internal expectations play a substantial role in memory as it would in perception.

Reply 

We would like to thank the reviewer for their exceptionally detailed, thoughtful, and helpful comments.

Reviewer 1 Comment 1

The authors have conducted a wide range of experiments that attempt to titrate this illusory effect but these experiments appear to vary in many critical ways (set size of memory array, proportion of real/transformed letters in the memory array, intervening arrays, delay interval to test, fixation cross cueing potential target, number of possible choices), making it difficult to follow the guiding theoretical principles and evaluate what about the empirical effect has reliably been established. Despite this, this effect appears to appear fairly consistently across all experiments which leads me to believe it is likely a reliable effect. 

Reply

With the introduction of the Bayesian analysis we have also more clearly laid out the 3 core hypotheses of our paper, and how they are addressed in each experiment. 

Hypothesis 1 (Illusory memories occur in STM) 

Hypothesis 2 (These illusions are memory-driven) 

Hypothesis 3 (The illusory memories are driven by expectations about the world) 

With this structure, we hope to demonstrate clearly that these 3 hypotheses are in line with the results of each experiment, illustrating the reliability of the observed effect of expectations on short term memory. 

Reviewer 1 comment 2

Although I am favorable to the idea that top-down expectations can influence short-term memories (in many ways) and find this empirical effect particularly interesting, I am not certain whether this effect is best characterized as being a short-term memory illusion. To me, an illusion invokes the idea that the short-term memory representation has frequently and substantially been truly altered, but that has not been clearly demonstrated yet in this manuscript.

Reply

Thanks for pointing this out. We decided to use the term illusion because the term false memory encompasses a large array of memory phenomena. However, the term illusion might be equally diverse in its usage, we now realize. To clarify what we mean with the term memory illusion, we have added a clear definition/ demarcation in the introduction on page 3

“These experiments specifically focus on a subset of false memories, namely what we call “memory illusions”: cases were people report with high confidence to have seen something that was not presented to them, i.e., the illusory experience of a reliable and reportable memory. An example of this would be clearly remembering that you put your keys on the diningroom table, while in reality you left them on the side table by the front door. Focusing on memory illusions instead of general false memory reports allows a separation of such incorrect but seemingly true memories from memory lapses that are filled in by best guesses (such as when you have no clue where you put your keys, but you check out the dining room table, because you very often drop them there). In such memory lapses, the reliance on pre-existing knowledge is a useful strategy to consciously fill gaps in memory, but it does not reveal whether prior knowledge can actually shape the content of STM.”

Here in my review, I hope to have made comments that lead to better clarity of the research, and better connect the research to the wider visual working memory literature – with the aim of providing useful suggestions to the authors in improving their manuscript. I admit that I could not devote as much effort to conduct a thorough review as I would have hoped, and apologize as a result if this review causes any unintended frustrations.

MAJOR COMMENTS

Reviewer 1 comment 3

I would like to compliment the authors for committing to sharing these materials on the Open Science Framework, which will better archive the scientific outputs and promote the transparency and reproducibility of the research. However, I wished I could have informed this present review with access to the experimental code and data, and analysis code. One way I would have informed my review with access to open data would be to better understand the proportion of high-confidence responses being made as they vary across the many conditions, and attempt to reproduce the main statistical analyses – I was missing clear reporting on the proportion and variation of high-confidence responses made by each participant, which I think is critical for interpreting these results.

Reply

All raw data and the JASP analysis files for the 4 experiments reported in this paper can be found on OSF: DOI 10.17605/OSF.IO/DFBVA. In addition, we now report in the methods > analysis of each experiment the mean, range, minimum and maximum of the percentage of trials that were high confidence responses.

Reviewer 1 comment 4

Taking Experiment 1 for example, participants completed 240 trials in total. As there was only one pseudo-letter in the memory array for these experiments, there were only 40 trials where these were tested, and participants would make a high-confidence response on a smaller proportion of that. Further, there were three delay intervals employed (150, 350 and 750 ms) further partitioning the number of trials. Thus, I am concerned about whether any tested effect is a reliable and consistent empirical phenomenon and whether this effect is unduly influenced by different bias across participants to respond with overall high-confidence on the 4-point scale. 

Reply 

Based on this comment, as well as the comments of the editor, we have taken a critical look at the number of trials included in the central analyses of each experiment. As Reviewer 1 correctly points out, particularly in experiment 1 an extremely limited number of trials was part of the analysis. Therefore, we have decided to remove this experiment from the article, as the limited number of trials made it unreliable. Moreover, we now only include participants in the final analysis if they have at least 10 high confidence trials in each cell of the central analysis. Notably, this does not change the results of the 4 experiments reported in this manuscript. Both the stricter exclusion criterion and the transition to a fully Bayesian analysis lead to the same main effects and interactions and overall conclusions for the current version of the manuscript, compared the initial version that we submitted. 

We want to stress that we did not remove the original Experiment 1 because the results changed when we removed participants with too few trials, it was simply impossible to analyze the results of the original Experiment 1, because based on the new participant exclusion rule we would have excluded ALL participants. 

Reviewer 1 comment 5

One initial worry is that perhaps individuals that selectively report high-confidence responses will generally make a substantial contribution to the reported effects (as proportions of errors will be overall larger values). Thus, the illusory effect in the manuscript may be mostly determined by an overall overconfidence in memories rather than specific confusions regarding the memory that is driving their response.

Reply

The choice to only analyse high-confidence responses does mean that for some individuals we include more trials than for other individuals. However, our crucial tests are about the effects of target type and memory delay within these high confidence responses. If an individual is at a general level biased towards or away from high confidence in their confidence ratings, this bias is likely to take place independently of the type of target, or of the memory delay. Any effects that we observe of target type or memory delay are therefore showing up in spite of individual differences in meta-cognition bias. 

Reviewer 1 comment 6

I wanted to note that this effect does not seem contingent on high-confidence responses – confusions of pseudo-letters for real letters still occurred for low-confidence responses albeit at a lower rate (35% compared to 22%, as reported in Experiment 2, p.27). I do appreciate that the confusion rate is higher with high-confidence responses, a perhaps surprising trend, but with a bit of uncertainty around these estimates, I’m uncertain on how to best draw an inference. I think it would be worthwhile to present in the manuscript how these responses change as a function of confidence across all experiments to show that there is an illusory effect that happens moreso with high-confidence responses.

Reply 

The relevant difference between high and low confidence responses is that for the low confidence responses we can observe a clear pattern of guessing: participants are much more likely to choose a response option complete unrelated to the target (what we in the manuscript refer to as ‘other errors’) in low confidence responses than in high confidence trials (respectively 51% vs 10% in Experiment 1, 50% vs 11% in Experiment 2, 31% vs 3% in Experiment 3 and 46% vs 7% in Experiment 4). For the low confidence trials, the observed percentages of these other (non-illusion) errors are therefore in line with participants just randomly selecting one of the response options in the response display, especially in the high interference trials. This clearly indicates that the low confidence trials are not a relevant category to learn anything about the contents of memory: participants are (as they themselves also indicate by their low confidence ratings) not really remembering anything and just respond randomly. This is why we do not focus on low confidence trials in the analyses. We therefore feel that making comparisons between low and high confidence trials is not meaningful, as these two categories reflect completely different processes (guessing vs relying on information in memory).

We do mention the low confidence error rates in that section of the discussion, because with regards to response biases the low confidence trials become relevant: if people have no memory, it would make sense that their behavior is guided by response biases, but we don’t actually observe that. This makes it less likely that there are response biases at play in the high confidence trials. 

Reviewer 1 comment 7

The increase in memory illusions over time (from short delay to long delay) was interpreted to mean that “correctly encoded percepts of a pseudo-letter are being converted to illusory memories of a real letter” (p. 46). One could suggest that the reason that this effect is not bi-directional (real letters being converted to pseudo-letters) is because familiar, real letters are better maintained, resistant to interference or confusion. There is evidence that familiarity, or real-word stimuli such as letters (apologies for the self-citation), are afforded better processing into and consolidation within working memory:

Ngiam, W. X., Khaw, K. L., Holcombe, A. O., & Goodbourn, P. T. (2019). Visual working memory for letters varies with familiarity but not complexity. Journal of Experimental Psychology: Learning, Memory, and Cognition, 45(10), 1761.

Brady, T. F., Störmer, V. S., & Alvarez, G. A. (2016). Working memory is not fixed-capacity: More active storage capacity for real-world objects than for simple stimuli. Proceedings of the National Academy of Sciences, 113(27), 7459-7464.

Jackson, M. C., & Raymond, J. E. (2008). Familiarity enhances visual working memory for faces. Journal of Experimental Psychology: Human Perception and Performance, 34(3), 556.

Xie, W., & Zhang, W. (2017). Familiarity increases the number of remembered Pokémon in visual short-term memory. Memory & cognition, 45(4), 677-689.

Asp, I. E., Störmer, V. S., & Brady, T. F. (2021). Greater visual working memory capacity for visually matched stimuli when they are perceived as meaningful. Journal of Cognitive Neuroscience, 33(5), 902-918.

Hedayati, S., O’Donnell, R. E., & Wyble, B. (2022). A model of working memory for latent representations. Nature Human Behaviour, 6(5), 709-719.

Brady, T. F., & Störmer, V. S. (2022). The role of meaning in visual working memory: Real-world objects, but not simple features, benefit from deeper processing. Journal of Experimental Psychology: Learning, Memory, and Cognition, 48(7), 942.

Reply

Thanks for this helpful overview, we have added the references to the section on strength of the memory representation on page 38: 

An alternative to the idea that STM illusions are based on predictive processing, could be that the strong priors strengthen the encoding of the memory representation and make it less susceptible to decay. Indeed well-known real-life categories are linked to a very strong, well established representation in memory while less know stimuli like pseudo-letters are not (Adolph & Kretch, 2015; Asp, Störmer, & Brady, 2021; Brady, Störmer, & Alvarez, 2016; Brady & Störmer, 2022; Hedayati, O’Donnell, & Wyble, 2022; Ngiam, Khaw, Holcombe, & Goodbourn, 2019; Xie & Zhang, 2017)

However, we also argue that if pseudoletters are less solidly consolidated in memory, one should expect another pattern of results than we currently observe, namely that:

[partipants] should be selecting all items in the probe display with equal probability, as the representation of the target has been lost. This should result in similar rates of errors (selecting probe items that are not the target, or a mirrored version of the target) and illusions (selecting the mirrored version of the target). Our results show that this is not the case: Illusory memories are much more common than errors when the target was a pseudo-letter. Moreover, the incidence of expectation induced illusory memories increases as the memory trace weakens, and this is not (or rather, much less) the case for general errors.

Reviewer 1 comment 8

I appreciate the framework that the authors describe in the ‘The origin of illusory memories in STM’ section in the discussion (p. 47-50) – that top-down effects can be produced by predictive processing or protection from decay. To me, this entails differences in either encoding into short-term memory stores or consolidation within the store. I wonder if the authors could better specify their theory or the mechanism by which the illusion is happening here, as to better clarify their results.

Therefore, I am unsure whether the authors can make a strong case for illusory memories here, when the effect may be best described as a function of both guessing and familiarity benefits. One experiment design that I think would be informative in disentangling real-world experience of stimuli and current situational beliefs about the stimuli (say, the proportion of real-world letters and transformed letters in the arrays) would be reversing the proportion of transformed letters to real-world letters (more transformed letters than real letters) – this should manipulate the participant’s expectations about whether the target is more likely a pseudo-letter and you might see a ‘reversal’ of the illusory effect shown here, and a better specification of the effect.

Reply 

Experiment 2 shows that tacit patterns in the stimuli (cue colour indicating the likelihood of a pseudo-letter target) do not influence our observed effects. However, none of the experiments manipulates pseudo-letter likelihood at a more explicit level. This is an interesting suggestion attempting to disentangle encoding accounts (where one would expect that even when pseudo-letters occur more often, the well-known letter category would be better encoded and more resistant to memory illusions) vs predictive processing accounts (the internal prediction that the target is most likely a pseudo-letter would lead to more real-to-pseudo errors than pseudo-to-real errors). Replacing life-long priors (this is what the letters of your alphabet look like), with a new local prior (in this experiment, the F is always presented in a mirrored fashion) most likely requires extensive training. However, even after such training, it will be extremely difficult to really know how strong the newly acquired prior is compared to the life-long prior. And without knowing whether the local prior is stronger than the life-long acquired prior, a finding that after learning a local prior participants still show the standard pseudo-to-real dominance in memory illusions could also simply be indicative of the dominance of letter categories as a prior vis a vis the newly acquired prior.

With this, we are hesitant to present this experiment as a watertight follow up experiment (although we are curious, so we might run it anyway at some point just to see how it turns out)

Reviewer 1 comment 9

In Experiment 3, the color of the fixation cross acted as a cue – red indicated the pseudo-letter was the target on 25% of trials, blue indicated the pseudo-letter was a target on 60% of trials. The authors report the participants were not consciously aware of this pattern, before collapsing across these conditions in subsequent analyses. Could the authors indicate how this was tested, such as whether the participants were explicitly tested on these percentages, or whether they noticed any changes in likelihood of the pseudo-letter being a target?

Reply

On page 21 we have added 

“Participants filled out a brief questionnaire in which they were invited to indicate what the experiment was about after the experiment. The relationship between the colour of the cue and the nature of the target was not mentioned in any of the responses.” 

Reviewer 1 comment 10

In addition, was there any error correction applied for any post-hoc comparisons? For example, such as with the pairwise comparisons between the various conditions in Experiment 2 (short+interference versus long, short versus long), or comparing within high-confidence trials to within low-confidence trials. If there was any error correction applied, it would be best to explicitly mention that in the manuscript.

Reply

With the introduction of the full Bayesian analysis we do not run NHST follow up analyses, so multiple comparisons corrections are no longer relevant. 

Reviewer 1 comment 11

I think the Bayes Factors in the manuscript might need a check through, perhaps using the BF10 or BF01 notation to indicate the direction of the likelihood ratio.

On page 15, there is a typo in reporting the Bayes Factor in text – “However, there [was] no significant interaction between error type and time (F(2,106) = 1.68, p = .19, eta^2 = 0.031, BF = 0.131). The Bayes Factor of .31 indicates substantial evidence for the Null hypothesis.”

On page 45, the second paragraph, the Bayes Factor reported is 2.2, but it is described in-text as ‘only showed anecdotal evidence for the absence of an effect’. If this BF is consistent with how it is reported in the manuscript, this would be evidence in favor of the alternative hypothesis, rather than for the absence of the effect.

I think the Bayes Factor should be reported for all analyses in the manuscript, rather than left to the Appendix. I think the way it is reported presently in the manuscript suggests or encourages that the Bayesian analysis only be employed to evaluate evidence for the null hypothesis. While an advantage of Bayesian analyses is that it can address inferential issues around null hypothesis significance testing by providing likelihood ratios for competing hypotheses, it should not be used to selectively evidence particular results such as the null hypothesis. I do see how reporting all the results may crowd the results paragraph, so I leave this up to the authors’ discretion on how they report the values.

Reply

All experiments have now been updated to include a full Bayesian analysis (and the datasets on which the analyses are done have changed because of the change in exclusion criteria), so all BF values have been updated throughout the paper. The two examples of BF’s with a typo / incorrect interpretation are no longer part of the manuscript. 

Reviewer 1 comment 12

On page 42, last paragraph, there is a typo in – “As in Experiment 5, Hypothesis 1…” – I believe the authors meant Experiment 4 here.

Reply

Thanks for pointing this out, with the removal of experiment 1, we have had to redo the experiment numbering throughout the paper, including this reference. 

Reviewer 1 comment 13

 The term ‘psuedo-letter’ is used throughout the manuscript, but I think I prefer the term ‘transformed’ or in later experiments ‘mirrored letter’. There exists non-letter character sets like the Brussels Artificial Character Sets, which are artificial letters so as to better distinguish non-letter stimuli, letter-like stimuli and the manipulations in the present experiment.

Reply

Thank you for this suggestion, which we considered carefully. After consideration, we decided to stay with the term ‘pseudo-letter’ since it conveys that the characters retain some letter-like properties without being actual letters/graphemes. Moreover, our work was partially inspired by the influential manuscript of de Gardelle, Sackur and Kouider (2009) entitled “perceptual illusions in brief visual presentations”. In this paper they attempted to show that especially the early, unattended, stages of visual memory are sensitive to memory illusions (which they call perceptual illusions). They assert that working memory (i.e. attended visual memory) should be less susceptible to such illusions. Here we present data that finds the opposite of their claim. Crucially, throughout the paper they refer to mirrored or rotated letters as pseudo-letters. Since our design is inspired by this paper, this provides us with an additional motivation to keep our nomenclature in line with theirs. 

Reviewer 1 comment 14

The y-axis in all figures are labelled as ‘percentage of trials’, which I found confusing to parse. I initially interpreted this as percentage of total trials in the experiment, rather than a proportion of ‘high-confidence trials’ for example.

Reply

We agree with the reviewer that percentage of trials is actually a pretty uninformative label, thanks for pointing that out. The Y-axis has been relabeled on all figures to “error percentage”. We have considered following the reviewers suggestion to change it to % error in high confidence trials, but decided against this because the label was long and felt superfluous. With the new set up of the manuscript, we a) from the start clarify that we only analyse the high confidence trials, and b) only include figures that represent that subset of the data. The original experiment 1 (which was deleted) also included a figure which represented all data (and not just the high confidence data). With the removal of this experiment from the manuscript, we felt that there was enough coherence and clarity about the high confidence trials that we could stick to a slightly more concise y-axis label. 

Reviewer 1 comment 15

This effect reminds me of ‘swap errors’ and ‘feature misbinding’ errors that have been documented in the visual working memory literature. I think a potentially fruitful thread would be to consider whether the ‘identity’ of the letters are apprehended faster or better encoded than the ‘transformation’ of the letters – it might be worthwhile considering a mechanism whereby the retrieval of pseudo-letters may be via encoding the transformation of the letter anchored to the upright real letter (and that this transformation is decayed). I’ve attached some references here where there has been debate as to what generates these kinds of errors, whether they are guesses or a function of feature variability.

Bays, P. M. (2016). Evaluating and excluding swap errors in analogue tests of working memory. Scientific reports, 6(1), 1-14.

Fougnie, D., & Alvarez, G. A. (2011). Object features fail independently in visual working memory: Evidence for a probabilistic feature-store model. Journal of vision, 11(12), 3-3.

Pratte, M. S. (2019). Swap errors in spatial working memory are guesses. Psychonomic bulletin & review, 26(3), 958-966.

McMaster, J. M., Tomić, I., Schneegans, S., & Bays, P. M. (2022). Swap errors in visual working memory are fully explained by cue-feature variability. Cognitive psychology, 137, 101493.

Reply

This is indeed very interesting. We did notice that Pratte (2019) shows that these types of swap errors occurs almost exclusively in low confidence responses, which suggest that this process is less likely to be involved in our results. We have integrated this suggestion on page 19:

This also shows that the current illusory memories are quite different from earlier reports of swap errors in STM (Bays, 2016; Fougnie & Alvarez, 2011; McMaster, Tomić, Schneegans, & Bays, 2022; Pratte, 2019), where participants incorrectly report having seen features of a distractor, instead of the target feature. While the currently observed memory illusions are more likely to occur in high confidence trials than on low confidence trials, swap errors generally occur on low confidence trials instead of high confidence trials (Pratte, 2019).

William Xiang Quan Ngiam

I sign all my reviews, regardless of the recommendation to the editor. By signing this review, I affirm that I have made my best attempt to be polite and respectful while providing criticism and feedback that is hopefully helpful and reasonable.

Thank you! We greatly appreciate your constructive comments and admirable attitude to reviewing. We hope our responses and changes to the paper do justice to the time you have kindly put in.

Reviewer 2 Comments

In the present manuscript, the authors perform five behavioral experiments to evaluate “illusions” in short term memory, and whether they are perceptual effects or if they arise in working memory. Specifically, participants remember 6-8 letters, a subset of which are “pseudo” (mirrored) letters. After a cue and some delay, participants are probed to report the cued letter. Participants are more likely to mistake a pseudo letter for its real counterpart than they are to mistake a real letter for its pseudo letter counterpart, and this effect is more pronounced with increasing delay. As such, the authors conclude that short term memory illusions are not perceptual errors but arise in memory.

I think the premise of this paper is interesting and that the experiments are thorough. I particularly appreciated the inclusion of experiments 4 and 5, as I found myself very much stuck on the imbalanced memory and probe displays in experiments 1-3 and was glad to see those concerns addressed here. I do have some remaining concerns, which I detail below in (mostly) the order that they arose.

Reviewer 2 comment 1

I did have somewhat of a hard time with the use of the word illusion in this context. As someone who is admittedly more of a vision researcher, to me, the canonical definition of an illusion feels necessarily perceptual (i.e., where visual illusions arise in low-level visual areas). It seems to me that “illusion” is a term used in long-term memory research to be synonymous with false memories, and that in this context it’s meant to reflect false shorter term memories rather than an illusion in the (what I consider) traditional sense. My sense is that I won’t be the only reader with this hang-up, but that it could be quelled early in the manuscript by more clearly introducing the concept of illusions in long term memory and what exactly is meant by illusions in this context.

Reply

We initially decided to use the term illusion because the term false memory encompasses a large array of memory phenomena. However, the term illusion might be equally diverse in its usage, we now realize. To clarify what we mean with the term memory illusion, we have added a clear definition/ demarcation in the introduction, as indicated in the reply to Editor comment 1, as well as in our reply to Reviewer 1.

Reviewer 2 comment 2

De Gardelle et al (2009) is quite relevant to the current work and should be more clearly detailed when it is first introduced. For example, on my first read through it wasn’t clear to me what the “real counterpart of the rotated letter” meant, and how that was relevant to the central theme of this paper.

Reply

We have adapted the description of the De Gardelle study to clearly introduce the mirrored pseudo-letters with an example: 

“For example, De Gardelle and colleagues (De Gardelle, Sackur, & Kouider, 2009) showed participants displays which sometimes contained rotated letters (i.e., a pseudo-letter such as Ɔ). The participants, however, only indicated seeing the real counterpart of the rotated letter (i.e., they report seeing C).”

Reviewer 2 comment 3

What are the trial counts in the critical conditions in the experiments? In experiment 1, there are only ~300 trials, and ~100 in each of the delay conditions. It’s not reported how many of these are high confidence trials, but only ~10% of high confidence trials in each condition are meaningful errors, which means there are < 10 trials per cell. 

Reply

Because of this very valid concern about the number of trials included in the analysis, we have removed experiment 1 from this manuscript (see also the reply to Editor comment 2 and Reviewer 1 comment 4)

Reviewer 2 comment 4

Relatedly, the sample sizes change quite dramatically across some of the experiments. It’s particularly large for experiment 3, and the justification for this wasn’t clear. 

Reply

There are indeed rather large differences in sample sizes between the experiments. We now explicitly address how these sample sizes were determined for experiment 2-4 (experiment 1 was a convenience sample).

The very large sample of experiment 2 was part of a big research credit programme in which all first years participated, so the number of participants was not controlled by us. We make this explicit in the methods section of Experiment 2:

“The participants participated for student credits as part of a course within the first-year Psychology curriculum. The experiment was part of a 2-hour session which contained this experiment, and several other unrelated experiments. The number of participants was determined by student enrollment in the course.”

In experiment 3 and 4 we applied a Bayesian stopping criterium. We have included a description of this stopping criterion in the methods section of experiment 3 and refer to it in Experiment 4. 

“Note that in this experiment the number of participants was determined based on a stopping rule implemented in an ongoing sequential analysis of the Bayes Factor for the critical t-test, namely the comparison between the effect of Error-Type for Real and Pseudo-letter targets. Every other day of testing, the sequential analysis was run, and once the outcome crossed a BF of 10, testing would stop.”

Reviewer 2 comment 4

Did the authors run any power analyses to determine sample sizes/trial counts? And what was the stopping rule for data collection?

Reply 

We had a stopping rule in Experiments 3 and 4 that we now include in the methods section as can be seen in our reply to comment 3 above and also in the reply to Editor comment 3

Reviewer 2 comment 5

Related to the above, it would be helpful to see the breakdown of high vs. low confidence responses. The critical analyses only include high confidence responses, but it’s not reported what proportion of trials were high confidence.

Reply

We now include information about the proportion of high confidence trials in the methods section of each experiment (see pages 13, 22, 28 and 32).

Reviewer 2 comment 6

Experiment 2: It’s not clear why the proportions of target/response options was changed here. If the target is a pseudo-letter, its counterpart (a memory lure) is presented as a response option 50% of the time, but if the target is a real letter, it’s pseudo version (memory lure) only appears as a response option 25% of the time. Doesn’t that make them more likely to make a pseudo to real error on a higher proportion of trials than a counterpart to pseudo error? After seeing experiments 4 and 5, I’m mostly convinced that these regularities aren’t sufficient to cause the critical patterns, but I think the choice to change these proportions in experiment 2 should be more clearly motivated (why was this change made?).

Reply

Thanks for pointing out that the description of the trial-type distribution in experiment 1 (formerly experiment 2) was not optimally described. We have adjusted the methods section to more clearly lay out what the participants saw, and which of these trials were critical to our research question. It is important to note that in absolute numbers, the critical trials were equally distributed over pseudo-to-real and real-to-pseudo trials. However, in the experiment as a whole (including filler trials that were irrelevant for our research question) participants were more likely to encounter real letters as a target and as a correct response option. Our revised text is included below (which now appears on p. 11)

The target letter was a real letter in 75% of the trials and a pseudo-letter in the remaining 25%. If the target was a pseudo-letter, then it appeared as a response option randomly on half of those trials while on the other half of pseudo-letter trials only the real counterpart was one of the response options. If the target was a real letter, then it appeared as a response option on a randomly selected 83% of the trials, while its mirror reversed (pseudo-letter) version appeared as a response option on the other 16% of the trials. This ensured that of the critical trials (trials where participants could exhibit memory illusion: trials in which a real- or pseudo-letter target was combined with its mirrored counterpart in the response screen) about half were real-to-pseudo trials (approximately 13% of all trials), while the other half were pseudo-to-real trials (again, approximately 13% of all trials). Together, the critical trials made up ~220 trials, 26% of all trials in the experiment. It is important to note, though, that throughout the experiment participants were more likely to encounter real letters as the target, and as the correct response option, than pseudo-letters, in line with the overall expectation that real letters are the standard letter-form.

Reviewer 2 comment 7

The logic behind including the short with interference condition is a bit unclear. Was this just to make the task more difficult/what was the prediction?

Reply 

The goal of both the memory delay and visual interference condition was to diminish the strength of the memory representation. The prediction was that with more memory decay the relative strength of internal predictions would increase, and thus lead to more memory illusions. We have adjusted the relevant prediction of (the new) Experiment 1 to clarify this point more on page 8. 

2) These reports reflect illusory memories, and are not the result of incorrect perceptual encoding: the high confidence illusory reports become more prevalent as memory deteriorates, through the passage of time passes (Jacob, Breitmeyer, & Treviño, 2013) or trough interference (Ricker, Vergauwe, & Cowan, 2016)

Reviewer 2 comment 8

I wonder about the contributions of verbal vs. visual working memory to overall performance in this task and, more importantly, to the systematic bias in response errors. Given that the authors are interested in comparing perceptual vs. memory-based errors, it makes sense to use a task that is designed to rely on visual working memory. With stimuli like letters, however, it’s likely that participants are using the cue to rehearse the target letter using verbal working memory. Verbalizing the target in this way (i.e., rehearsing ‘B’ without rehearsing its pseudo vs. real letter classification) may contribute to the higher instance of pseudo letter to real counterpart illusions, particularly over time as memory starts to decay and reliance on verbal working memory increases. I appreciate that the data focus on high confidence errors, but I think this type of rehearsal lends itself to such errors. Is it possible that the specific pattern of errors reported here arise from a reliance on verbal working memory? If so, does that affect the interpretation of the data, or might it simply be the case that the mechanism underlying these illusions is increased reliance on more course information in memory, such as category information rather than a precise visual representation? I would be curious to see the experiment done with a concurrent articulatory suppression task. I don’t think this is necessarily required, but I am curious to hear the authors’ thoughts on this, either in a response or in the manuscript.

Reply

This is a very interesting suggestion, which we have included in the general discussion on page 40:

FTT does suggest that pseudo-letters have a verbal (the letter itself) and a gist level (“mirrored”) representation, while real letters can be stored purely at the verbal level. This would mean that verbal rehearsal is more effective to store a complete representation of real letters in working memory, but not for pseudo-letters. Removing verbal rehearsal as a strategy, for example through articulatory suppression, would therefore impact the storage of the verbal representation in working memory, leading to less stable memory representation of real letters. It is worthwhile to test, in future research, whether memory traces for real letters which cannot rely on verbal rehearsal are similarly sensitive to memory illusions (real-to-pseudo) as pseudo-letter representations are in our experiments. This would clarify whether the current effects are the result of maintenance effects (real letters are more effectively maintained than pseudo-letters) or prediction-related effects at retrieval (strong internal priors are more likely to influence ambiguous memory representations).

Minor:

Reviewer 2 comment 9

- Figure 3: It wasn’t immediately clear to me the importance of the dotted line at 20% in the left panel. Also, for this and subsequent similar graphs, you might consider changing the label on the x axis to Time from array offset (rather than time to offset, which I found a little confusing).

Reply

With the removal of Experiment 1, this figure has also been removed from the paper, and subsequent figures did not include this dotted line. In addition, for all figures, the x-axis is now labeled Memory Delay, in line with the name of the relevant factor within the design of experiments 1 to 4. 

Reviewer 2 comment 10

- Page 18 mistakenly says “occurred either appear”

- Page 15 – there was no significant interaction between error type and time (missing word “was”)

Reply 

These errors have been corrected, thanks for pointing them out.

Reviewer 2 comment 11

- Regarding the way timing is indicated on the trial schematics: rather than 0.75 – 3 s, it might be clearer to label them.75 or 3 s

Reply

We have implemented this change in figure 1 and figure 3

Reviewer 2 comment 12

- Was there really a 1.5 to 3.75 s delay following the probe on trials where the probe and irrelevant letter display were sequential as well? In-text suggests that the second display onset 5 s after the offset of the first letter display, but wouldn’t this depend on whether the trial belonged to the short or long delay condition? If this labeling is correct, then in the 3 s condition, the delay between offset of the first letter display and onset of the second could be over 7 seconds.

Reply

Indeed, as mentioned in the text, the irrelevant letter display appeared 5 seconds after the memory display. Since the probe appeared either shortly after the memory display, or after a longer period of time, the interval between probe and irrelevant letter display varied accordingly, so that in sum the total interval was always 5 seconds. We have elucidated this in the text in the following passage: 

If the memory probe appeared in isolation (the Short and Long conditions), the irrelevant second display appeared 5 seconds after memory display offset, which meant that the interval between the probe and the irrelevant second display was either presented for 1.5s (Long delay) or 3.75s (short delay). 

Reviewer 2 comment 13

- Some of the language regarding support for the various hypotheses is a bit strong throughout. Specifically, language that describes the hypotheses as “confirmed” rather than, for example, supported.

Reply 

We have replaced confirmed with supported in sentences related to hypothesis testing.

- Were the letters in the first and second letter displays (the memory array and the interference display) selected randomly with or without replacement? (Could letters reappear in the interference display?)

Reply

Indeed letters were selected with replacement, so they could appear both in the memory and the interference display.

Reviewer 2 comment 14

- The intro says there are four experiments but the paper includes 5

Reply

Since we have removed experiment 1 from the paper, we had to go over the entire manuscript to change the overall number of experiments from 5 to 4, which makes this initially erroneous mention of 4 experiment ultimately the correct reference.

---

## [Decision Letter · Decision Letter 1]

6 Mar 2023

Seeing Ɔ, remembering C: Illusions in short-term memory

PONE-D-22-23654R1

Dear Dr. Otten,

We’re pleased to inform you that your manuscript has been judged scientifically suitable for publication and will be formally accepted for publication once it meets all outstanding technical requirements.

Kind regards,

David K Sewell

Academic Editor

PLOS ONE

Additional Editor Comments (optional):

Dear Dr Otten,

Thank you very much for revising your original submission. I want to thank you for the care you've taken in preparing your revision. The focus on the "core 3" hypotheses has helped provide a more salient throughline connecting the experiments and the additional descriptions you've provided regarding theoretical constructs and methodological choices have made the manuscript clearer to follow.

I have now heard back from both of the original reviewers (Reviewer 1 is William Ngiam). Both of the reviewers are satisfied that you have addressed all of their concerns. Reviewer 1 has several other recommendations, but notes that these are entirely voluntary. Given the PLOS ONE publication criteria have been met, my disposition is to accept the revision as is, but leave you the option to address the additional points raised by Reviewer 1 as you see fit.

Congratulations on this very interesting work! I want to thank you and the reviewers for your time and efforts.

Yours sincerely,

David K Sewell

Reviewers' comments:

Reviewer's Responses to Questions

**Comments to the Author**

1. If the authors have adequately addressed your comments raised in a previous round of review and you feel that this manuscript is now acceptable for publication, you may indicate that here to bypass the “Comments to the Author” section, enter your conflict of interest statement in the “Confidential to Editor” section, and submit your "Accept" recommendation.

Reviewer #1: All comments have been addressed

Reviewer #2: All comments have been addressed

2. Is the manuscript technically sound, and do the data support the conclusions?

Reviewer #1: Yes

Reviewer #2: Yes

3. Has the statistical analysis been performed appropriately and rigorously? 

Reviewer #1: Yes

Reviewer #2: I Don't Know

4. Have the authors made all data underlying the findings in their manuscript fully available?

Reviewer #1: Yes

Reviewer #2: Yes

5. Is the manuscript presented in an intelligible fashion and written in standard English?

Reviewer #1: Yes

Reviewer #2: Yes

6. Review Comments to the Author

Reviewer #1: The authors were diligent in their response to the comments on the manuscript, making substantial changes accordingly. I have a few minor comments for the authors’ consideration, which do not require any additional changes to the manuscript – they are written with the hope that it may improve their manuscript further.

1. I commend the authors for uploading the raw data and JASP analysis files for all experiments reported in the paper to an OSF repository – I consider this one hallmark of rigorous science. I accessed the raw data, but found it difficult to initially parse the column values. I suggest that the authors include a README file to detail how the raw Excel data files are organized.

I cross-referenced the statistical values from the shared JASP analysis files with the manuscript and found they were consistent throughout. I also re-analysed the data from Experiment 1 without the short with interference condition (only short versus long) in the memory delay because I wasn’t sure if it was appropriate to include those conditions together as a single variable – if a significant result was found, would it be attributed to an effect of interference or an effect of delay duration? Nevertheless, I found the pattern of results was unchanged – that there is credible evidence for a main effect of delay duration – consistent with the other experiments in the manuscript.

2. I realized in this round of review that it is probably appropriate to cite some work that has tied confidence ratings to the likelihood of remembering an item, and the precision of the memory representation. Some VWM theorists tie the variability in confidence directly to the noise of the working memory representation (precision estimates in delayed recall tests), advocating that the uncertainty is part of the representation. Notably, Adam and Vogel (2017) find this correlation but find a positive bias – participants tend to be overconfident and often fail to detect lapses. (I should have remembered this literature in my last review, and I’m sorry to again suggest references in my review).

Rademaker, R. L., Tredway, C. H., & Tong, F. (2012). Introspective judgments predict the precision and likelihood of successful maintenance of visual working memory. Journal of vision, 12(13), 21-21.

Van den Berg, R., Yoo, A. H., & Ma, W. J. (2017). Fechner’s law in metacognition: A quantitative model of visual working memory confidence. Psychological review, 124(2), 197.

Adam, K. C., & Vogel, E. K. (2017). Confident failures: Lapses of working memory reveal a metacognitive blind spot. Attention, Perception, & Psychophysics, 79, 1506-1523.

3. Given the previous point, I have been perseverating on the third hypothesis – does real world knowledge drive STM illusions? In support of this hypothesis, the authors have observed pseudo-letter targets are confused for their real counterparts more than the vice versa, within high-confidence responses. I agree that this effect is likely driven by real-world knowledge in some way but I do have a remaining concern that this effect may be situational. Can other expectations or other factors influence one’s priors or expectations that would override the influence of real-world knowledge here?

In the response to my initial comments on this matter, the author’s highlighted (the now) Experiment 2 – a manipulation of the fixation cross color as a cue for low- and high-probability trials had no effect on the pattern of results. However, the participants also reported no awareness of this pattern. This leaves open the possibility that if participants have explicit awareness of the statistical regularities (the cueing of target in this case), then this manipulation would induce differences in the pattern of results, perhaps a reversal of the effect – that the cue may now drive the types of memory illusions (high-confidence target reversals) from real letter to pseudo-letter.

I think it may help to consider what role the real-world knowledge plays here in the memory illusions. Is the real-world knowledge perhaps driving an overconfidence in metacognition, in tandem with higher quality memories? I think the other reviewer made a good point – that the identity of the target may have been verbally rehearsed, whereas the ‘gist’ (whether the letter was mirrored) was susceptible to decay. (The author’s do discuss this on page 42, paragraph 2). So, in a way, the participant may respond overconfidently because they can intuit they are very good at maintaining the real letter identity, but not realize how susceptible one is to losing the gist (a shift in criterion). That is to say, perhaps it is worthwhile considering the memory illusion effect here as playing on the positive bias to trust one’s memory and failure to detect lapses.

Anyhow, I agree with the authors that this concern alone does not warrant running a follow-up experiment like the one I had mentioned in my previous review (reversing the proportion of pseudo-letters and real-world letters beyond half-and-half). I included these comments here in case the authors’ find the thread useful for an addition to their manuscript or future follow-up experiments.

Thank you for recognizing the effort in my previous review, and responding conscientiously to my previous comments.

William Xiang Quan Ngiam

I sign all my reviews, regardless of the recommendation to the editor. By signing this review, I affirm that I have made my best attempt to be polite and respectful while providing criticism and feedback that is hopefully helpful and reasonable.

Reviewer #2: The authors have addressed all of my comments. They indicate that all materials are available, but I wasn't able to access them using the link/information provided.

7. PLOS authors have the option to publish the peer review history of their article (what does this mean?). If published, this will include your full peer review and any attached files.

Reviewer #1: **Yes: **William Xiang Quan Ngiam

Reviewer #2: No

---

## [Editor Report · Acceptance letter]

13 Mar 2023

PONE-D-22-23654R1 

Seeing Ɔ, remembering C: Illusions in short-term memory 

Dear Dr. Otten:

I'm pleased to inform you that your manuscript has been deemed suitable for publication in PLOS ONE. Congratulations! Your manuscript is now with our production department. 

Kind regards, 

on behalf of

Dr. David Keisuke Sewell 

Academic Editor

PLOS ONE